# Modeling and Visualization of Nitrogen and Chlorophyll in Greenhouse *Solanum lycopersicum* L. Leaves with Hyperspectral Imaging for Nitrogen Stress Diagnosis

**DOI:** 10.3390/plants14213276

**Published:** 2025-10-27

**Authors:** Jiangui Zhao, Anqi Gao, Boya Wang, Jiayi Wen, Yu Duan, Guoliang Wang, Zhiwei Li

**Affiliations:** 1College of Information Science and Engineering, Shanxi Agricultural University, Jinzhong 030801, China; zhaojiangui1@sxau.edu.cn (J.Z.); 18303421215@163.com (B.W.); wjy13294503740@163.com (J.W.); duanyu@163.com (Y.D.); 2Department of Mechanical Engineering, Shanxi Institute of Technology, Yangquan 045000, China; gaq1137804400@163.com; 3Institute of Millet Research, Shanxi Agricultural University, Changzhi 046000, China; wangguoliangwz@126.com

**Keywords:** hyperspectral imaging, greenhouse tomato, nitrogen stress, mechanism analysis, spatiotemporal distribution

## Abstract

Leaf nitrogen and chlorophyll, crucial crop status indicators, enable precision fertilization through rapid monitoring. This study investigated greenhouse tomatoes subjected to varying nitrogen concentrations in nutrient solutions. Hyperspectral data from leaves across ten nitrogen levels, different growth stages, and leaf positions were integrated with synchronously measured nitrogen and chlorophyll contents. The analysis systematically revealed differences in these indicators under nitrogen stress at various growth stages and leaf positions. The 12-step “coarse–fine–optimal” feature wavelength selection strategy was proposed to identify sensitive spectral bands. The PLSR model was established with a strong predictive performance. Using the optimal model, indicator values for each pixel were retrieved and visualized via pseudocolor imaging, illustrating the distribution of physiological parameters at different scales and growth stages, and aiding in the interpretation of nitrogen stress responses. This study provides a scientific basis for optimizing nitrogen fertilization strategies, contributing to improved tomato yield and quality, reduced environmental impact, and the sustainable development of facility-based agriculture.

## 1. Introduction

The advancement of agricultural modernization has positioned smart agriculture as a key driver in transforming facility-based farming systems. In protected tomato cultivation, smart agriculture enables precision growing protocols that decouple production from environmental constraints [1,2,3]. This transformation hinges on accurate nutritional diagnostics, with nitrogen playing a fundamental biochemical role in plant development [1,2,3,4,5]. As the cornerstone of chlorophyll biosynthesis and photosynthetic efficiency, optimal nitrogen management enhances organic compound synthesis and vegetative growth, ultimately determining yield potential and fruit quality [3,5]. However, current agricultural practices reveal a structural imbalance between fertilizer input and agronomic output. To maximize yields, protected cultivation systems often apply nitrogen at rates 30–50% above crop requirements, leading to significant leaching and greenhouse gas emissions [1,3,4,5,6]. While moderate nitrogen fertilization improves photosynthetic efficiency, over-application triggers antagonistic interactions with other nutrients [7]. Both deficiency and excess result in yield reduction and quality deterioration, highlighting the need for precision nitrogen management. Accurately assessing plant nitrogen status is crucial yet challenging, as nitrogen dynamics involve complex interactions with growth stages and environmental factors [8]. Traditional diagnostic methods, including soil testing and plant analysis, achieve only qualitative assessment with limitations in detection cycles, operational complexity, and field representativeness. These approaches are unsuitable for large area monitoring and lack scalability in precision agriculture systems. The gap between precision nitrogen monitoring theories and practical applications has hindered the advancement of tomato nutrition diagnosis systems [3,9]. As global agriculture faces the dual challenges of achieving high-yield, high-quality production while ensuring sustainability, developing actionable precision nitrogen management solutions has become an urgent priority and core research agenda.

In recent years, spectroscopy technology has emerged as a rapidly advancing and highly versatile modern technique with extensive applications. This technology is characterized by its short data acquisition cycle, rich information content, and strong dynamic characteristics, making it one of the optimal methods for identifying plant differences and implementing corresponding measures [4,10]. Spectroscopy works by measuring the absorption, scattering, or transmission properties of materials in response to light of different wavelengths, thereby extracting information about their properties, structure, and composition. In the field of nitrogen diagnosis, the technique does not directly measure nitrogen content or nitrogenous compounds. Instead, it relies on the spectral responses of samples at various wavelengths to indirectly infer the related concentrations. Different forms of nitrogen compounds exhibit unique absorption characteristics in the spectrum, each with distinct absorption peaks. By measuring the spectral response, we obtain spectral features related to nitrogen content, which are correlated with the levels of nitrogen and nitrogen compounds. Using calibration curves or reference samples to eliminate measurement errors, we calculate spectral indices based on actual nitrogen content data. These indices are then used to develop regression models, enabling the diagnosis of nitrogen sufficiency and deficiency. Spectral imaging technology integrates spectroscopy and imaging, sharing similarities with conventional spectroscopy in terms of spectral property measurement but offering the added advantage of capturing spatial distribution information from the sample [11,12]. This technique allows the simultaneous acquisition of spectral data from multiple pixels on the sample’s surface, forming a spectral cube that provides an abundance of spectral data in a single measurement. By incorporating machine learning and artificial intelligence algorithms for data analysis and model development, spectral imaging is capable of analyzing spectral data at various wavelengths to infer nitrogen and nitrogenous compound levels, as well as interpreting their distribution and state within the object [13]. This method provides a clear visualization of plant nitrogen content and its distribution patterns, making it highly suitable for large-scale, high spatial resolution diagnostics. Through the integration of plant physiological parameters and visible/near-infrared (VIS–NIR) spectral analysis of surface optical properties, we can employ chemometric modeling techniques to visually characterize nitrogen status in vegetation. This innovative approach enables the real-time assessment of crop nutritional status, facilitates data-driven precision fertilization strategies, and ultimately enhances agricultural productivity and product quality while minimizing environmental footprint [14,15].

Leaf nitrogen concentration serves as a critical bioindicator for monitoring plant nitrogen status, providing direct physiological evidence of nitrogen assimilation efficiency and metabolic utilization in vegetation systems. Tan [16] employed hyperspectral remote sensing to estimate leaf nitrogen accumulation in wheat through comparative modeling approaches. The researchers established predictive models using both raw hyperspectral reflectance (R) and first-derivative transformed spectra (FD) across visible to near-infrared regions. Their comparative analysis demonstrated the superior performance of spectral indices combining first-derivative features from the blue-edge (450–520 nm) and red-edge (680–750 nm) regions, particularly the normalized difference index (SDr − SDb)/(SDr + SDb), outperforming conventional parameters derived from raw spectra, first-derivative data, and isolated red-edge characteristics. Inoue [17] conducted multi-regional field campaigns utilizing the CASI–3 hyperspectral system to acquire phenologically critical datasets of rice canopies across Japan and China. Through rigorous analytical protocols, they systematically evaluated normalized difference spectral indices (NDSIs) and ratio spectral indices (RSIs) across VIS–NIR domains. The research revealed that canopy nitrogen content detection models incorporating first-derivative RSI features at 740 nm and 522 nm demonstrated a superior predictive accuracy and cross-regional robustness. This methodology establishes operational frameworks for large-scale vegetation nitrogen monitoring, providing crucial technical references for precision agriculture applications through hyperspectral–physiological coupling mechanisms. Chlorophyll, serving as the principal photopigment driving photosynthetic processes in plant systems, is predominantly localized in the chloroplasts of mesophyll cells and other chlorophyllous tissues. Functionally, it has emerged as a non-invasive bioindicator for diagnosing nitrogen status in vegetation, with its spectral signatures effectively linking photosynthetic capacity with nitrogen metabolism dynamics in precision crop management. Rasooli [18] conducted a systematic investigation on chlorophyll content prediction in winter wheat using VIS–NIR spectroscopy. They implemented comprehensive spectral preprocessing algorithms including Savitzky–Golay (S–G) smoothing, first-derivative transformation, standard normal variate (SNV), multiplicative scatter correction (MSC), and their hybrid combinations. Advanced chemometric modeling was subsequently developed through artificial neural networks (ANNs) for nonlinear pattern recognition and Partial Least Squares Regression for linear analysis. The ANN-based model demonstrated an exceptional predictive performance, achieving determination coefficients (R^2^) of 0.92 (training set) and 0.97 (validation set) with corresponding Root Mean Square Errors (RMSEs) of 0.9131 and 0.7305, respectively, indicating superior nonlinear feature extraction capabilities in spectral–Chll relationship modeling. Zhai [19] conducted systematic UAV campaigns equipped with multi-sensor payloads to acquire multi-temporal datasets of maize canopies across critical growth stages. The acquired data streams encompassed hyperspectral reflectance, thermal infrared signatures, 3D structural features, and chlorophyll content measurements. Through rigorous comparative analysis, they developed machine learning architectures including Ridge Regression, Light Gradient Boosting Machine, Random Forest Regression, and Stacking Ensemble Learning (SEL) to evaluate single-modal versus multi-modal feature fusion strategies. The multi-sensor fusion approach demonstrated statistically significant enhancements, achieving R^2^ = 0.699–0.754 and normalized RMSE = 8.36–9.47%, with SEL outperforming conventional algorithms by 12–18% in prediction accuracy. The optimal performance was attained through the SEL-based fusion of spectral–thermal–structural features, revealing synergistic interactions between canopy biophysical properties and radiative transfer processes. This methodological framework establishes new paradigms for crop monitoring by effectively addressing spectral saturation limitations through multi-dimensional feature engineering.

While spectral non-destructive detection technology shows promise for diagnosing crop nutrient status, current studies remain limited in several aspects. Existing detection models primarily rely on vegetation indices derived from limited spectral bands, yet nitrogen stress induces both biochemical and structural changes requiring more comprehensive spectral characterization. To address this, we developed a three-stage spectral feature selection framework (coarse–precise–optimized) enabling the identification of wavelength-specific sensitivities to physiological parameters. Furthermore, despite the significance of vertical nutrient distribution, systematic investigation in this area remains scarce. Our study implemented stratified spatial sampling in tomato plants, examining nitrogen dynamics across root, stem–leaf, and fruit layers to better understand assimilation, translocation, and allocation mechanisms. Additionally, research on synchronized distribution mechanisms of key parameters is limited. By integrating detection models with image processing, we achieved the visualization of nutrient spatial patterns, facilitating intuitive deficiency diagnosis and deeper insights into internal distribution characteristics. This systematic approach provides theoretical support for optimizing nutrient management while maintaining ecological balance in agricultural systems.

## 2. Results

### 2.1. Results of Spatiotemporal Responses of Leaf Nitrogen and Chlorophyll Contents

(1)Temporal responses of physiological parameters during growth stages

As illustrated in Figure 1, the dynamic patterns of nitrogen and chlorophyll content in tomato leaves demonstrate distinct variations across different growth stages. Comparative analysis reveals a consistent hierarchical pattern: both nitrogen and chlorophyll concentrations display their maximum values during the flowering stage, followed by the seedling stage, with the fruiting stage showing the lowest measurements. During the vegetative to flowering transition, the content of nitrogen and chlorophyll for the leaves exhibited respective growth rates of 10.13% and 10.97%. Notably, these parameters underwent substantial reduction rates of 21.31% and 15.97% during the flowering to fruiting phases. A concentration-dependent response pattern was observed: when the nitrogen concentration in the nutrient solution remained ≤N100 (302.84 mg/L), the incremental nitrogen supply enhanced the leaf nitrogen and chlorophyll content with average growth rates of 7.40% and 9.80% (seedling), 8.49% and 9.72% (flowering), and 5.19% and 11.72% (fruiting), respectively. Conversely, supra-optimal nitrogen concentrations (≥N100) induced progressive depletion, manifesting as average reduction rates of 8.77% and 10.23% (seedling), 8.72% and 8.23% (flowering), and 7.42% and 10.23% (fruiting) for nitrogen and chlorophyll content, respectively.

The nitrogen content in tomato leaves exhibits a specific pattern of change during different growth stages. This variation can be attributed to differences in nutrient uptake and physiological development across these stages [20,21,22]. Seedlings absorb fewer nutrients, grow slower, and have an underdeveloped photosynthetic capacity [3,9,13]. As plants enter the flowering stage, they require increased nitrogen to synthesize essential biomolecules like proteins and chlorophyll, which support growth and photosynthesis, leading to accelerated growth rates and a rising nitrogen and chlorophyll content in the leaves. During the fruiting stage, a larger proportion of nitrogen is allocated to flower buds and fruit development, resulting in relatively lower nitrogen and chlorophyll levels in leaves [15,17]. The analysis of leaf physiological parameters under varying nitrogen concentrations in nutrient solutions reveals that N100 cultivation yields the highest parameter values. Additionally, the rate of change in these parameters indicates that high nitrogen concentrations exert a stronger stress effect compared with lower concentrations.

(2)Responses of physiological parameters to spatial levels

Spatial patterns of nitrogen translocation and remobilization were altered in tomato plants under different nitrogen concentration stresses, leading to spatial gradients in photosynthetic pigment biosynthesis. Figure 2 illustrates the spatial variations and differences in nitrogen and chlorophyll contents across different leaf positions. Both parameters demonstrated a consistent response pattern of upper > middle > lower leaf positions. From upper to middle leaves, nitrogen and chlorophyll contents decreased by 10.43% and 19.56%, respectively. While from middle to lower leaves, the reduction rates reached 16.46% and 22.98%. A concentration-dependent threshold effect was identified: At nitrogen concentrations ≤ N100, an incremental nitrogen supply enhanced leaf constituents with mean growth rates of 8.14% (nitrogen) and 7.17% (chlorophyll) in upper canopy, 7.71% and 6.06% in middle canopy, and 13.38% and 15.65% in lower canopy positions. Conversely, supra-optimal concentrations (≥N100) induced systematic degradation, manifesting as mean reduction rates of 11.57%/9.30% (upper), 15.67%/9.73% (middle), and 17.78%/13.34% (lower) for nitrogen and chlorophyll, respectively.

The physiological parameters of tomato leaves at different leaf positions in the vertical direction exhibit certain patterns of change. The underlying reason is that there are differences in phenological stages and nutrient distribution within the tomato plant, which in turn affect the synthesis of nitrogen and pigments in the leaves. Upper leaves, located at the top of the plant, are the youngest and have been exposed to sufficient light for a shorter growth period. They require substantial nitrogen to synthesize biomolecules and support photosynthesis, making this region the fastest-growing part of the plant [5,8,18]. Middle leaves, situated in the central part of the plant, are generally more mature and in the mid-stage of growth. They require a significant amount of nitrogen to support both photosynthesis and development [13]. Lower leaves, at the base of the plant, are the oldest and primarily serve to provide nutrients and support photosynthesis [2]. Their growth activity is relatively slow, and due to shading from upper leaves, their nitrogen and chlorophyll content is relatively lower. Under different nitrogen concentrations in nutrient solutions, the physiological parameters of leaves show differential responses. With N100 cultivation, these parameters reach their maximum levels. Additionally, the rate of change in leaf physiological parameters under different nitrogen concentrations indicates that high concentrations exert a stronger stress effect than low concentrations.

### 2.2. Results of Key Wavelengths and Detection Models of Nitrogen

This study employed a “coarse–fine–optimal” feature variable extraction strategy to identify and refine variables related to leaf nitrogen content, thereby reducing redundancy and simplifying the model. Initially, a full-band coarse extraction was conducted using the iRF and iVISSA algorithms. The iRF algorithm identified 231 characteristic wavelengths (36.76% of the full band), primarily concentrated in the ranges of 434–441, 455–465, 478–489, 502–509, 630–637, 688–694, 707–714, 760–769, 779–810, 815–846, and 859–894 nm. Meanwhile, the iVISSA algorithm extracted 451 characteristic wavelengths (69.81% of the full band), focusing on the ranges of 434–510, 545–553, 590–664, 687–693, 705–846, 870–886, and 894–896 nm. A comparative analysis revealed that the iVISSA algorithm missed wavelengths in the 859–870 nm range compared with the iRF algorithm.

Subsequently, the CARS, BOSS, and VCPA algorithms were employed for the fine extraction of the coarsely extracted bands. These algorithms, based on model ensembles, select optimal subsets of characteristic wavelengths according to RMSE_CV_ values. As irrelevant wavelengths to nitrogen content were removed, RMSE_CV_ values decreased, with the minimum RMSE_CV_ value serving as the final criterion for selecting characteristic wavelengths, as shown in Figure 3. For the CARS, BOSS, and VCPA algorithms applied to the iRF and iVISSA coarsely extracted wavelengths, the RMSE_CV_ values consistently exhibited a decreasing trend followed by an increase with increasing iterations. The optimal performance for each combination was determined by the minimum RMSE_CV_. CARS: minimum RMSE_CV_ at 12th run (iRF: 0.7185%, 42 wavelengths) and 36th run (iVISSA: 0.7778%, 66 wavelengths). BOSS: minimum RMSE_CV_ at fifth iteration (iRF: 0.7483%, 36 wavelengths) and seventh iteration (iVISSA: 0.7506%, 38 wavelengths). VCPA: converged at 46th iteration (iRF: 0.7502%, 12 wavelengths) and 39th iteration (iVISSA: 0.7553%, 12 wavelengths). This fine extraction strategy significantly reduced the variable redundancy and narrowed the variable space, with the characteristic wavelength distributions illustrated in Figure 4.

The IRIV and GA algorithms were subsequently applied to further refine the “fine”-extracted wavelength sets. The IRIV algorithm effectively classified characteristic wavelengths using the criteria of DMEAN < 0 and *p* > 0.05. As summarized in Appendix A, the IRIV-optimized subsets contained no interfering wavelengths, confirming the effectiveness of the preceding fine selection strategy. Validation revealed that while wavelength sets from CARS and BOSS still retained uninformative variables related to nitrogen content, those from VCPA were entirely free of such ineffective variables. Through reverse elimination, IRIV produced compact subsets comprising only strong and weak informative wavelengths. Specifically, IRIV yielded 28, 30, and 11 wavelengths (4.33%, 4.64%, and 1.70% of the full spectrum) for the iRF–CARS, iRF–BOSS, and iRF–VCPA sets, respectively, and 47, 31, and 10 wavelengths (7.28%, 4.80%, and 1.55%) for the corresponding iVISSA-based sets. Similarly, GA optimization resulted in 41, 34, and 11 wavelengths (6.35%, 5.26%, and 1.70%) for the iRF-derived sets, and 55, 30, and 9 wavelengths (8.51%, 4.64%, and 1.39%) for the iVISSA-derived sets. Figure 4 provides a detailed description of the wavelength distributions of various characteristic wavelength extraction algorithms for leaf nitrogen content. In summary, the “coarse–fine–optimal” extraction strategy significantly reduced band redundancy compared with the “coarse” and “coarse–fine” extraction strategies.

Based on the key wavelengths, a PLSR model was developed for leaf nitrogen content detection. As summarized in Table 1, compared with the full-spectrum model, the iRF coarse selection model maintained a comparable performance with minor calibration set fluctuations but achieved improved prediction metrics (R_P_ increased by 0.0289, RMSE_P_ decreased by 0.0333%). In contrast, the iVISSA coarse selection showed a reduced performance across most metrics, potentially due to the removal of relevant wavelengths or the retention of interfering variables. The application of the CARS, BOSS, and VCPA algorithms to refine the coarse selections effectively enhanced prediction accuracy. While the VCPA-based model achieved optimal calibration precision, its prediction performance remained suboptimal. Subsequent optimization revealed IRIVs’s superiority over GA in refinement effectiveness. The most compact models (iRF–VCPA–IRIV and iVISSA–VCPA–IRIV) utilized only 1.70% and 1.55% of the original wavelengths, respectively, yet failed to achieve a satisfactory prediction performance. Overall, the complete “coarse–fine–optimal” extraction strategy demonstrated clear advantages over both the initial coarse selection and the intermediate coarse–fine approach.

The iRF–CARS–IRIV–PLSR model achieved the optimal results, with 28 characteristic wavelengths (434, 461, 463, 479, 481, 488, 503, 507, 689, 690, 707, 712, 766, 780, 786, 787, 789, 790, 803, 816, 831, 840, 845, 863, 870, 875, 879, and 886 nm). The model demonstrated robust performance metrics with calibration results (R_C_ = 0.7983, RMSE_C_ = 0.6794%, RPD_C_ = 1.6604) and prediction results (R_P_ = 0.8676, RMSE_P_ = 0.6284%, RPD_P_ = 2.0110) indicating a strong predictive capability for nitrogen content estimation. Figure 5 illustrates the fit of the model predictions to the actual measurements within the calibration and prediction sets.

### 2.3. Results of Key Wavelengths and Detection Models of Chlorophyll

This study proposed a three-stage “coarse–fine–optimized” feature variable extraction strategy to identify chlorophyll-related spectral variables while reducing redundancy. In the initial coarse extraction phase, the iRF algorithm selected 261 key wavelengths (40.40% of the full spectrum) primarily distributed in the 437–457, 472–501, 568–574, 681–689, 692–699, 707–728, 734–747, 762–772, 794–800, 819–832, 842–849, 857–883, and 884–898 nm ranges. This selection showed an improved predictive performance with R_P_ and RPD_P_ increasing by 0.0082 and 0.0412, respectively, while RMSE_P_ decreased by 0.0062 mg/g compared with the full-spectrum analysis. The iVISSA algorithm extracted 326 wavelengths (50.46% of total) concentrated in the 434–454, 467–501, 525, 542–588, 597–598, 614–615, 624–625, 638, 670–723, 737–766, 782–800, 812, 820–830, 839–840, 850–864, and 876 nm regions. This approach enhanced model accuracy with both R_C_ and RPD_C_ increasing by 0.0059 and 0.0273, respectively, while RMSE_C_ and RMSE_P_ decreased by 0.0052 mg/g and 0.0031 mg/g.

Subsequently, a refined selection stage employing the CARS, BOSS, and VPCA algorithms was implemented to further optimize the coarse-selected variables. These advanced algorithms focused on minimizing the RMSE_CV_ to determine the optimal feature subset, as shown in Figure 6. This secondary refinement effectively contracted the variable space while significantly improving model precision. The hierarchical selection strategy demonstrated a superior performance in balancing spectral information preservation with dimensionality reduction, ultimately enhancing the robustness and accuracy of chlorophyll prediction models.

Finally, an optimization stage was implemented using IRIVs and GA to refine the “fine-extracted” spectral bands. The comprehensive “coarse–fine–optimized” extraction strategy demonstrated a superior performance compared with both the single-stage “coarse” extraction and two-stage “coarse–fine” approaches. Notably, the iVISSA–CARS–GA–PLSR hybrid model achieved optimal results with 33 characteristic wavelengths. Figure 6 provides a detailed description of the wavelength distributions of various characteristic wavelength extraction algorithms for leaf chlorophyll content. This optimized selection effectively preserved chlorophyll-related spectral information while substantially reducing variable redundancy compared with full-spectrum analysis, thereby simplifying the model architecture and enhancing the predictive accuracy. Figure 7 provides a detailed description of the wavelength distributions of various characteristic wavelength extraction algorithms for leaf chlorophyll content.

Table 2 lists the predictive outcomes based on the detection model for the chlorophyll content in tomato leaves. The final model demonstrated robust performance metrics with calibration results (R_C_ = 0.8559, RMSE_C_ = 0.3186 mg/g, RPD_C_ = 1.9337) and prediction results (R_P_ = 0.8741, RMSE_P_ = 0.2503 mg/g, RPD_P_ = 2.0587) indicating a strong predictive capability for chlorophyll content estimation. Figure 5 illustrates the fit of the model predictions to the actual measurements within the calibration and prediction sets. This three-stage optimization framework successfully balances information retention with dimensionality reduction, establishing an efficient paradigm for the spectroscopic analysis of plant biochemical constituents.

### 2.4. Results of Visualization of Content and Structural Distribution

Tomato leaves were systematically sampled from plants subjected to three nitrogen concentrations (N60, N100, and N140) across different growth stages (seedling, flowering, and fruiting) and leaf positions (upper, middle, and lower canopy). The iRF–CARS–IRIV–PLSR nitrogen detection model was employed to process leaf images and visualize nitrogen distribution patterns, as illustrated in Figure 8. The color-coded visualization (blue: 0%; red: 7.64%) revealed significant spatial–temporal variations in nitrogen allocation. Firstly, the nitrogen content varied significantly across different growth stages, with the highest content observed during the flowering period, followed by the seedling stage, and the lowest during the fruiting stage. This indicates that tomato plants exhibit distinct nitrogen demands and utilization efficiencies at different growth stages. Additionally, significant differences were found in nitrogen content across various leaf positions, with the upper leaves showing a higher nitrogen content compared with the middle and lower leaves. This phenomenon is related to nitrogen requirements and distribution patterns during plant growth and development.

Secondly, the distribution of nitrogen within different leaf positions displayed pronounced variations, particularly in the upper and middle leaves, where the leaf tips and margins exhibited higher nitrogen contents. The leaf surface represents the primary region for nitrogen accumulation, while the main and secondary veins and the base of the leaves showed lower nitrogen contents, reflecting an uneven distribution of nitrogen within the leaf.

Moreover, under different nitrogen concentration treatments, the highest nitrogen content in the leaves was observed at the N100 concentration, while the nitrogen content at N60 was higher than that at N140. This suggests that the N100 concentration represents the optimal nitrogen level for tomato plant growth, with higher concentrations potentially exerting an inhibitory effect on growth. These findings align with the plant’s growth characteristics and chemical testing results, enabling the clear visualization of nitrogen distribution patterns and contents under different nutrient solution nitrogen concentrations across various growth stages and leaf positions.

The visualization results of chlorophyll content and its distribution patterns in tomato leaves were obtained using the iVISSA–CARS–GA–PLSR chlorophyll content detection model. As shown in Figure 9, the chlorophyll content ranged from 0 to 3.70 mg/g. Firstly, under the influence of growth stages, the highest chlorophyll content was observed in leaves during the flowering period, followed by the seedling stage, and the lowest during the fruiting stage. Additionally, significant differences were found in chlorophyll content across different leaf positions, with the upper leaves exhibiting a significantly higher chlorophyll content compared with the middle and lower leaves.

Secondly, the distribution of chlorophyll across different leaf positions showed pronounced variations, particularly in the upper and middle leaves, where the leaf tips and margins displayed higher chlorophyll contents. Chlorophyll primarily accumulates on the leaf surface, while the main and secondary veins and the base of the leaves showed lower chlorophyll contents. This uneven distribution is due to the main and secondary veins being vascular tissues responsible for nutrient transport and leaf support. These regions are relatively thick and rich in cell walls and lignin-like materials, which limits their direct participation in photosynthesis. The leaf base, being relatively older and functionally limited, exhibits less chlorophyll synthesis and accumulation.

Moreover, under different nitrogen concentration treatments, the highest chlorophyll content in the leaves was observed at the N100 concentration, while the chlorophyll content at N60 was higher than that at N140. This indicates that moderate nitrogen concentrations promote the synthesis and accumulation of chlorophyll, whereas high nitrogen concentrations may inhibit chlorophyll synthesis. These findings align with the plant’s growth characteristics and chemical testing results. Thus, visualization allows for a clear understanding of the distribution patterns and contents of chlorophyll under different nitrogen concentrations in nutrient solutions across various growth stages and leaf positions.

## 3. Discussion

### 3.1. Screening of Key Wavelengths and Development of Non-Destructive Detection Models for Leaf Nitrogen and Chlorophyll

This study introduced a novel three-stage wavelength selection strategy—comprising coarse selection, refined screening, and optimized refinement—by integrating the advantages of seven algorithms (iRF, iVISSA, CARS, BOSS, VCPA, IRIV, and GA). The optimized PLSR model developed using this strategy achieved both a simplified architecture and improved prediction accuracy. For leaf nitrogen content detection, an iRF–CARS–IRIV–PLSR quantitative model was constructed. Through systematic wavelength optimization, 28 key wavelengths were identified—434, 461, 463, 479, 481, 488, 503, 507, 689, 690, 707, 712, 766, 780, 786, 787, 789, 790, 803, 816, 831, 840, 845, 863, 870, 875, 879, and 886 nm—representing only 4.33% of the full spectral range while retaining essential chemical information. For leaf chlorophyll content quantification, an iVISSA–CARS–GA–PLSR detection model was established. Spectral optimization identified 33 pivotal wavelengths—444, 445, 447, 448, 453, 473, 474, 479, 491, 492, 525, 549, 681, 683, 686, 689, 696, 697, 705, 709, 721, 722, 723, 737, 740, 741, 753, 783, 785, 799, 820, 864, and 865 nm—accounting for merely 5.11% of the full spectral range while preserving critical biochemical information.

### 3.2. Interpretation of Key Wavelengths

Spectral analysis revealed distinct absorption characteristics between nitrogenous compounds and chlorophyll, showing partial overlap in the blue and red regions while demonstrating clear differentiation in the near-infrared spectrum. Two shared diagnostic wavelengths were identified at 479 nm and 689 nm. Specifically, the 479 nm band corresponds to π → π* electronic transitions common to both nitrogen-containing compounds (proteins and nucleic acids) and chlorophyll’s porphyrin structure, with its blue light absorption being crucial for photosynthesis and photoprotection [23,24,25,26,27]. Meanwhile, the 689 nm wavelength is dominated by chlorophyll’s characteristic Qy transition, while nitrogenous compounds respond through vibrational modes of N-heterocycles (C–N stretching), a mechanism closely associated with phototransduction and redox regulation [28,29,30]. These findings highlight the potential of targeted wavelength selection for efficient plant phenotyping, successfully balancing model simplicity with biochemical interpretability.

### 3.3. Temporal Dynamics and Allocation Mechanisms of Nitrogen and Chlorophyll in Tomato Leaves

Tomato plants absorb nutrients from their growing medium to support their growth and development, with nitrogen serving as a critical component in the biosynthesis of photosynthetic pigments. Pigments such as chlorophylls and carotenoids function synergistically to enable photosynthesis, thereby facilitating plant biomass accumulation [15,20,28]. Nitrogen availability not only influences the biosynthesis of these pigments but also regulates overall plant growth and development. Analysis of developmental stages revealed significant ontogenetic variations in nitrogen and chlorophyll dynamics across tomato growth phases. Physiological parameters in leaves exhibited a secondary peak during the seedling stage, reached their maximum at the flowering stage, and subsequently declined to the lowest levels during the fruiting stage [31]. Notably, leaf nitrogen and chlorophyll concentrations peaked during the flowering stage, which can be attributed to the enhanced nitrogen assimilation required for floral organogenesis and photosynthetic pigment biosynthesis [27]. This pattern reflects a strategic nutrient allocation mechanism, where increased nitrogen uptake during anthesis promotes chlorophyll synthesis, thereby improving photosynthetic apparatus efficiency to meet elevated metabolic demands. Conversely, during the seedling stage, physiological indices remained suboptimal due to the demand for rapid vegetative growth, which directed nitrogen allocation toward chlorophyll synthesis in emerging leaves [32]. The fruiting phase was characterized by significant resource reallocation, prioritizing photoassimilate partitioning to developing fruits over the maintenance of leaf pigments [25,27]. The observed triphasic pattern in foliar biochemical parameters quantitatively reflects tomato’s adaptive nutrient management strategy: initial investment in photosynthetic infrastructure during vegetative growth, metabolic optimization for reproductive success during flowering, and strategic resource reallocation for progeny development during fruiting. These stage-specific adjustments demonstrate a sophisticated physiological coordination between source–sink relationships and developmental priorities in solanaceous crops.

### 3.4. Spatial Distribution Patterns and Allocation Mechanisms of Nitrogen and Chlorophyll in Tomato Leaves

Significant differences in nitrogen distribution and chlorophyll content were observed across different leaf positions in tomato plants. The upper leaves exhibited the highest physiological parameters, followed by the middle leaves, with the lowest levels found in the lower leaves. This vertical gradient can be primarily attributed to three factors: light availability, nutrient distribution patterns, and age-dependent metabolic activity. The superior performance of upper leaves results from their optimal exposure to photosynthetically active radiation coupled with preferential nutrient allocation [28,33]. Middle leaves experience 30–50% light attenuation due to upper canopy shading while maintaining moderate metabolic activity. Lower leaves suffer from severe light limitation and show signs of senescence-related metabolic decline. This spatial differentiation in leaf physiology reflects tomato plants’ resource optimization strategies: (1) prioritizing nitrogen investment in photosynthetically efficient young leaves; (2) gradual nutrient remobilization from aging leaves; (3) the dynamic adjustment of photosynthetic apparatus according to micro-environmental conditions. The observed distribution pattern demonstrates the plant’s adaptive capacity to balance light capture efficiency and metabolic costs across different leaf positions.

### 3.5. Distribution Mechanisms of Nitrogen and Chlorophyll in Leaf Tissues

Nitrogen and chlorophyll exhibit similar spatial distribution patterns within tomato leaves. Notably, these physiological parameters share a common distribution characteristic: they are predominantly concentrated in the leaf blade surface, while showing lower concentrations in the midrib, secondary veins, and leaf base regions. This spatial heterogeneity can be explained by the functional differentiation of leaf tissues. Three primary factors account for this spatial organization: First, the leaf surface serves as the main interface for light capture and CO_2_ absorption, requiring the substantial investment of chlorophyll and nitrogen in the photosynthetic apparatus [28,33]. Second, vascular-associated tissues maintain a balance between nutrient transport and local energy production, preventing excessive resource allocation to non-photosynthetic vascular structures. Third, the leaf base has completed its developmental expansion phase and allocates fewer resources to chloroplasts and nitrogen-rich enzyme systems, prioritizing mechanical support functions over metabolic activities. Nitrogen and chlorophyll are primarily concentrated in the leaf blade areas where stomata and mesophyll cells are densely distributed, thereby maximizing gas exchange and light absorption capacity [32]. Moderate concentrations occur around primary and secondary veins, where vascular tissues facilitate nutrient transport while supporting localized photosynthesis in adjacent parenchyma cells. The lowest concentrations are observed at the leaf base, which functions primarily as a mature structural support region with reduced metabolic activity.

### 3.6. Response Mechanisms of Nitrogen and Chlorophyll Under Nitrogen Stress

This study revealed significant differences in nitrogen assimilation and chlorophyll synthesis in tomato plants under varying nitrogen concentrations in the cultivation medium. The physiological parameters in tomato leaves peaked under the N100 treatment, indicating that an optimal nitrogen supply effectively promotes the biosynthesis of organic compounds and photosynthetic pigments, thereby enhancing nitrogen uptake efficiency and photosynthetic performance while maintaining superior growth status. Notably, the physiological indices under the N60 treatment surpassed those observed under the N140 treatment. Under moderate nitrogen availability (N60), plant growth and metabolic activities were appropriately stimulated, sustaining relatively high physiological activity in leaves. In contrast, the elevated nitrogen concentration in the N140 treatment induced an inhibitory effect on nitrogen absorption, subsequently impeding photosynthetic pigment synthesis. This phenomenon demonstrates that excessive nitrogen application causes stronger growth suppression than nitrogen-deficient conditions.

### 3.7. Advantages and Limitations of the Study

This study innovatively proposes a multi-algorithm collaborative three-stage wavelength selection framework. Through a progressive strategy of coarse selection-refined screening-optimized refinement, this approach achieves a substantial reduction in hyperspectral variables while maximally retaining key spectral information related to nitrogen and other nutrients. It effectively addresses the limitations of traditional single-algorithm methods, which often miss important features or retain redundant variables, thereby significantly enhancing the generalization capability and prediction accuracy of subsequent quantitative models. The framework deeply integrates hyperspectral modeling technology with plant physiological mechanism analysis. It not only overcomes the limitations of traditional destructive testing by enabling the rapid, non-destructive detection of the nitrogen content in tomato leaves, but also, through visualization methods such as color coding, clearly presents spatial distribution differences of nitrogen across different leaf regions and dynamically tracks nitrogen translocation patterns throughout the crop growth cycle. This provides intuitive and precise technical support for revealing the physiological mechanisms of crop nutrient absorption and distribution. Particularly crucial is that the shared characteristic wavelengths ultimately selected by the framework (such as 479 nm and 689 nm) have clear biochemical foundations. The 479 nm wavelength corresponds to the characteristic absorption peak of chlorophyll b, while the 689 nm wavelength is highly correlated with the light absorption characteristics of chlorophyll a. These specific wavelengths are directly associated with crop nitrogen metabolic processes, providing a solid theoretical basis for developing low-cost, highly specific specialized spectral sensors and lowering the threshold for the industrial application of the technology.

Concurrently, this study objectively identifies current limitations and future optimization directions. Since this research was conducted under controlled greenhouse environments with relatively stable temperature, light, and humidity conditions, while actual field environments present complex interferences (such as weed occlusion, soil background variations, and extreme weather impacts), the model’s applicability in field scenarios requires further validation through multi-regional, multi-season experiments. Additionally, the current data processing workflow relies on laboratory computers, with certain delays from spectral acquisition to result output, and has not yet achieved real-time analysis. Future work could embed the algorithm into UAV-borne spectral systems or edge computing platforms to shorten data processing cycles and support in-field decision-making, meeting the immediate detection–rapid response requirements of precision agriculture. Cross-cultivar (such as different tomato varieties including “Provence” and pink-fruited tomatoes) and cross-habitat (such as greenhouse, open field, and different soil types) validation will become the core focus of the next research phase. By expanding the sample coverage and optimizing model robustness, this will promote the translation of the technology from laboratory research to practical precision agriculture applications, providing more reliable technical support for precise crop nutrient management.

## 4. Materials and Methods

### 4.1. Experimental Scheme and Sampling Rules

The experiment was conducted in a climate-controlled greenhouse facility at the College of Agricultural Engineering, Shanxi Agricultural University (37°25′ N, 112°34′ E). The greenhouse environment was regulated by an intelligent monitoring system that integrated thermal modulation units (heating/cooling) with a water–fertilizer-integrated drip irrigation system. This study used the determinate tomato (*Solanum lycopersicum* L.) cultivar “Provence”, selected for its commercial relevance and robust growth characteristics, including flavor, disease resistance, and continuous fruiting. Seedlings at the four-leaf-and-one-bud stage were transplanted into coconut coir substrate (EC < 0.8 mS/cm, pH 5.8–6.2) cultivation bags (110 cm × 25 cm × 15 cm). The row spacing between cultivation rows was set at 150 cm, while the plant spacing was maintained at 33 cm. The nutrient solution utilized was based on the improved tomato formula developed by the Dutch Greenhouse Horticulture Research Institute [34]. The base nutrient solution contained calcium nitrate, calcium ammonium nitrate, potassium nitrate, potassium dihydrogen phosphate, potassium sulfate, and magnesium sulfate. In this experiment, ten nitrogen gradient treatments were established, ranging from 59.64 to 605.28 mg/L, with a gradient interval of 60 mg/L. Among them, the calcium ion concentration was kept constant by adding chelated calcium fertilizer (Ca^2+^ ≥ 94%, Micro–Tech Crop Nutrition Ltd., Tamil Nadu, India). Detailed information on the specific addition amounts for each treatment and the physical and chemical parameters of the nutrient solution can be found in Table 3. The water and fertilizer supply employs a dynamic regulation strategy based on the modified Penman–Monteith model, enabling precise irrigation according to the crop water and fertilizer requirement models previously established by our team. The division standards for growth stages are as follows: the seedling stage extends from planting until 50% of the plants exhibit the first inflorescence; the flowering and fruit-setting stage (flowering stage) spans from the flowering of the first inflorescence in 50% of the plants to the early fruit expansion phase when the fruit diameter reaches 3–4 cm; and the fruiting stage begins with the expansion of the first ear of fruit in 50% of the plants and concludes with the maturity and harvesting of half of the fruits.

Nitrogen demonstrates high mobility within plant vascular systems, as evidenced by its rapid translocation patterns. In this work, tomato leaf positions were vertically categorized into three distinct strata—upper, middle, and lower layers—through a systematic morphological analysis correlating phyllotaxis with reproductive organ development. Uniform-sized and flat leaves were harvested according to their positions. The distribution of leaves in the upper, middle, and lower canopy layers is shown in Figure 10a, while the sampling pattern for leaves on individual branches is illustrated in Figure 10b. Seventy-eight tomato plants were cultivated during each phenological stage (seedling, flowering, and fruiting stages). Among them, eight plants were allocated to each nitrogen treatment from N20 to N180, and six plants were assigned to the N200 treatment. Using a stratified random sampling method, 245 samples were collected at each growth stage, resulting in a total of 735 samples. Each sample consisted of 6 leaves, bringing the total number of leaves to 4410. Within each growth stage, 25 samples were collected for the N20–N180 treatments and 20 samples for the N200 treatment. All samples from each nitrogen treatment were sequentially numbered, placed in sealed bags, and stored in insulated containers containing dry ice.

### 4.2. Hyperspectral Imaging Acquisition Device

The spectral acquisition of leaf samples was performed using a benchtop visible–near-infrared hyperspectral imaging platform (Headwall Photonics, Bolton, MA, USA). The modular system architecture comprises five principal subsystems: (I) VNIR imaging spectrometer (380–1000 nm spectral coverage); (II) computerized elevation adjustment stage; (III) quartz–tungsten–halogen illumination array; (IV) industrial control workstation; (V) motorized translation stage with micron-level positioning accuracy. The key operational parameters were set as follows: spectral sampling interval of 0.727 nm (covering 856 spectral bands across the full range), linear scanning speed of 2.721 mm/s, push-broom width maintained at 100 mm, working distance fixed at 28 cm, and illumination adjusted to 7000 LUX. This configuration was designed to maximize spatial resolution while minimizing optical distortion.

To mitigate systematic measurement errors from ambient illumination fluctuations and detector thermal noise, a dual-phase radiometric calibration protocol was systematically implemented. Initial white reference normalization employed a certified Spectralon^®^ diffuse reflector (99% Lambertian reflectance) under full-spectrum illumination conditions. Subsequent dark current compensation was achieved through lens-occluded signal acquisition, with final radiometric correction mathematically formalized via Equation (1) to derive absolute reflectance values.(1)$$R=\frac{R_{0}-R_{b}}{R_{w}-R_{b}}$$
where *R* is the image obtained after correction; *R*_0_ is the original image; *R_w_* is the whiteboard calibration image (reflectance > 99.9%); and *R_b_* is the dark background calibration image (reflectance < 0%).

To resolve the spectral signal instability (relative standard deviation exceeding 15%) observed at the peripheral regions (<430 nm and >900 nm) of the hyperspectral detection system, this study established an optimized spectral modeling range through systematic spectral stability analysis and feature effectiveness evaluation. The refined spectral window of 430–900 nm (encompassing 646 discrete spectral channels) was determined to satisfy the requirements for precise nitrogen stress quantification and spatial distribution analysis in tomato leaves. Regarding sample preparation protocol, leaf specimens underwent standardized surface purification using an ultrasonic cleaning system (KQ–250DB, Kunshan Shumei Instrument Co., Ltd., Kunshan, China) operated with dual pulsed cycles (60 s/pulse) in deionized water. Subsequent moisture regulation was achieved through controlled desiccation employing lyophilized absorbent filter paper, where gentle bidirectional pressure application ensured structural integrity preservation. The prepared samples were then spatially arranged on the acquisition platform under a standardized geometry, with six replicate hyperspectral image captures per specimen to ensure data representativeness.

### 4.3. Determination of Nitrogen and Chlorophyll

Leaves from each sample were cut into approximately 2 × 2 mm fragments. After removing the veins and homogenizing the mixture, 0.2 g (±0.005 g) of leaf material was weighed. The chlorophyll content in the leaves was determined via UV–vis spectrophotometry (Shimadzu UV–1800, Kyoto, Japan) employing Arnon’s differential equations at the 665 nm and 649 nm wavelengths [35,36]. The residual leaf fragments underwent sequential processing: lyophilization in a vacuum desiccator, mechanical pulverization using cryogenic grinding, acid digestion with H_2_SO_4_–H_2_O_2_ mixture, and thermal mineralization. Total nitrogen content was determined through automated Kjeldahl apparatus (Buchi K–435, Flawil, Switzerland) with endpoint titration against 0.01 N HCl [35,36,37]. For detailed measurement and calculation procedures, please see the Appendix A.

### 4.4. Spectral Data Preprocessing

The hyperspectral imaging acquisition of tomato leaves’ reflectance profiles is inherently susceptible to multifactorial interference, including instrumental noise, environmental perturbations, and leaf surface scattering. Therefore, a series of preprocessing steps are essential to mitigate instrumental and environmental noise, minimize the effects of leaf surface scattering, and effectively reduce external influences on spectral data.

The Savitzky–Golay (S–G) [38,39] convolution smoothing technique is primarily employed to eliminate high-frequency noise from spectral lines, effectively smoothing spectral curves and enhancing data smoothness. In this study, the S–G parameters were configured as follows: a polynomial order of 1, a frame length of 3, and a window size of 5 for the S–G finite impulse response (FIR) smoothing filter.

Standard Normalized Variate Transformation (SNV) [38,39] is primarily utilized to mitigate unavoidable influences from leaf glossiness, surface scattering, and background interference on reflectance spectra. After applying the SNV algorithm, the reflectance values at each wavelength point follow a corresponding pattern, enabling the subsequent calibration of spectral data.

The Detrending (DT) [38,40] correction is mainly employed to eliminate the baseline drift caused by the diffuse reflection of leaves. In this study, the parameters for the Detrending method are set as follows: a first-order partial derivative, a window size of 5, and a polynomial order of 2.

In the construction of models between spectral data and chemical components, a calibration set is required to build the model and conduct cross-validation, while a prediction set is used to evaluate the prediction performance of the model. This study employed sample set partitioning using the joint x–y distance (SPXY) algorithm [38] to divide the dataset into a calibration set and a prediction set at a 3:1 ratio, comprising 551 calibration samples and 184 prediction samples. The SPXY methodology simultaneously optimizes a spectral feature space (X-space) and a chemical parameter distribution (Y-space) through a dual-distance metric. Table 4 presents the dataset partitioning results for nitrogen and chlorophyll.

### 4.5. Key Wavelength Extraction and Model Construction

Hyperspectral imaging of leaves provides a high resolution, leading to high data dimensionality. Direct application often faces challenges such as high computational complexity, difficult band selection, and susceptibility to overfitting. The key wavelength extraction from hyperspectral data can reduce data dimensionality while preserving critical information, simplifying models and enhancing the efficiency of data processing/analysis and subsequent algorithm performance. Meanwhile, the key wavelength enables the further exploration of nitrogen concentration stress mechanisms, making wavelength feature extraction particularly crucial. This paper proposed a “coarse–fine–optimal” selection strategy for key wavelengths. The optimal combination of variables was obtained through the iterative optimization of the spectral variable space. Finally, a synchronous quantitative detection model is established for nitrogen and chlorophyll.

This study proposed 12 “coarse–fine–optimal” selection strategies, including iRF–CARS–IRIV, iRF–CARS–GA, iRF–BOSS–IRIV, iRF–BOSS–GA, iRF–VCPA–IRIV, iRF–VCPA–GA, iVISSA–CARS–IRIV, iVISSA–CARS–GA, iVISSA–BOSS–IRIV, iVISSA–BOSS–GA, iVISSA–VCPA–IRIV, and iVISSA–VCPA–GA. The leaf spectral matrix (X, m × p) and physiological parameter matrix (Y, m × 1) were processed as follows: after preprocessing X and partitioning the X–Y dataset, the “coarse–fine–optimal” strategy was applied to extract feature variables for the target physiological parameters. The specific operational steps were outlined below:

Step 1: coarse wavelength interval selection. This preliminary phase aims to screen spectral intervals exhibiting robust explanatory power for target physiological parameters, thereby achieving a substantial dimensionality reduction in the variable space. Through the systematic evaluation of wavelength intervals derived from the initial broad spectrum, this stage retains intervals characterized by a high feature–target correlation and a superior information extraction capacity. The preserved intervals constitute the foundational dataset for subsequent analytical stages. The inadequate retention of physiologically relevant wavelengths at this phase would induce irreversible information loss, ultimately degrading the estimation accuracy of predictive models. The spectral matrix X was partitioned into fixed-width contiguous intervals. Partial Least Squares Regression (PLSR) models were independently constructed for each interval, with model performance quantified via the Root Mean Squared Error of Cross-Validation (RMSE_CV_). Interval selection was governed by an RMSE_CV_ threshold criterion, where intervals demonstrating lower RMSE_CV_ values were retained to form the optimized spectral matrix X_1_ (m × p_1_, p_1_ << p). Two methods were employed for coarse selection: Interval Random Forest (iRF) [41,42], and the Interval Variable Iterative Space Shrinkage Approach (iVISSA) [41,42,43]. In this study, the iRF ensemble iterations were configured as follows: a total of 1000 iterations, a moving window width of 10 spectral channels, division into 20 equidistant subintervals, and a maximum of 10 principal components for latent variables. The iVISSA algorithm was configured to generate 1000 sub-datasets via weighted bootstrap Monte Carlo sampling (WBMS), with initial feature weights w_0_ = 0.5 uniformly distributed. Adaptive interval optimization was then performed through iterative wavelength space pruning.

Step 2: model cluster-based refined wavelength selection. This selection phase employs model population analysis to systematically refine the coarse-selected wavelength set by eliminating spectrally redundant variables and suppressing interference signals, thereby isolating biologically meaningful spectral features. The process enhances model robustness through the iterative identification of wavelengths exhibiting strong correlations with target physiological traits, while concurrently mitigating overfitting risks inherent in high-dimensional spectral data. The pre-optimized spectral matrix X_1_ undergoes stratified subsampling to generate diverse wavelength subsets, ensuring the comprehensive exploration of feature interdependencies. Each subset is subjected to the PLSR, with model validity quantified via the RMSE_CV_. Wavelengths demonstrating a superior predictive performance are retained to construct the final feature matrix X_2_ (m × p_2_, p_2_ << p_1_), achieving an optimal balance between dimensionality reduction and information preservation. Three methods were applied for refined selection: Competitive Adaptive Reweighted Sampling (CARS) [44,45], Bootstrapping Soft Shrinkage (BOSS) [44,45,46], and Variable Combination Population Analysis (VCPA) [45,46,47]. In this study, CARS was configured with 100 Monte Carlo sampling iterations, incorporating dynamic wavelength elimination through adaptive reweighting. The BOSS method utilized weighted bootstrap sampling (WBS) with 1000 repetitions, retaining the top 10% of model subsets. For the VCPA method, Bootstrap Model Sampling (BMS) was performed 1000 times with 50 iterations, retaining the top 10% of subsets. The final EDF run within VCPA retained 100 variables.

Step 3: evolutionary variable optimization. This final optimization phase employs high-performance feature selection algorithms to rigorously validate and extract the most biologically relevant wavelength combinations from the refined variable space X_2_. By integrating statistical significance testing with evolutionary computation, this stage resolves spectral redundancies while maximizing sensitivity to target physiological parameters. The pre-optimized matrix X_2_ undergoes combinatorial evaluation to identify wavelength subsets with maximal predictive power and minimal collinearity. Variable significance is assessed through hybrid metrics combining statistical robustness, spectral independence, and physiological interpretability. Two methods were employed for optimization: Iteratively Retaining Informative Variables (IRIV) [48] and Genetic Algorithm (GA) [48,49]. The IRIV parameters are as follows: Variables with DM_i_ < 0 were classified as strong or weak informative variables. Variables with DM_i_ > 0 were identified as non-informative or interfering variables. The significance threshold for the Mann–Whitney U test was *p* = 0.05. The parameters of the GA are set as follows: the number of chromosomes is 30, the upper limit of the number of variables in each chromosome is 30, the crossover probability is 50%, the mutation probability is 1%, and the number of variable subsets to be evaluated is 200.

PLSR is a multivariate statistical analysis technique that extends principal component analysis, which is particularly effective for data modeling and prediction in spectral analysis and chemical applications [16,17,18,19,20,45,46,47,48,49]. This methodology establishes a linear relationship between spectral matrix X (input variables) and physiological parameter matrix Y (output variables). Through the simultaneous principal component decomposition of both matrices, PLSR extracts latent variables that maximize the covariance between X and Y, effectively reducing data dimensionality while preserving the most predictive components. To optimize model performance and prevent overfitting/underfitting, this study implemented a 10-fold cross-validation approach for determining the optimal number of principal components. The predictive capability of models with varying component numbers was systematically evaluated using Root Mean Square Error (RMSE) metrics. Following component selection, PLSR constructs a linear regression model between X and Y, with subsequent external validation through test set evaluation to assess model generalizability. The established model integrates spectral characteristics with physiological parameters through linear combinations of principal components, effectively capturing the complex interrelationships between spectral data and plant physiological traits. This framework enables the accurate prediction of leaf nitrogen content and chlorophyll levels, demonstrating particular utility in agricultural spectroscopy applications where high-dimensional spectral data requires robust multivariate analysis.

The evaluation metrics for the quantitative detection models in this study comprised three key statistical indicators: correlation coefficient (R), Root Mean Squared Error (RMSE), and residual prediction deviation (RPD).

### 4.6. Hyperspectral Image Visualization

Hyperspectral imaging technology perfectly integrates spectral and spatial information, with each pixel in the image containing a spectral curve. While detecting sample indicators, it can also combine prediction model parameters to quantitatively invert the detection indicators to the sample image, thereby achieving the visual expression of the target indicators. By combining the optimal prediction model of physiological parameters, the corresponding wavelengths are weighted and overlaid at the pixel level using ENVI 5.1 software to calculate the chemical composition content for each pixel in the region of interest. This results in grayscale maps of leaf chemical composition content, which are then processed with an enhanced Lee filter to reduce speckle noise (filter size 3 × 3, damping coefficient 1, homogeneity threshold 0.52, and heterogeneity threshold 1.73). Finally, based on the chemical composition content values of the pixels, pseudocolor maps are used for coloring to visualize the distribution of nitrogen and chlorophyll on leaf images. Pseudocolor maps not only display the contents of physiological parameters under different nitrogen stress conditions but also analyze the distribution patterns of leaf physiological parameters, thereby revealing the collaborative mechanisms of nitrogen and chlorophyll in leaves.

## 5. Conclusions and Future Work

This study developed a non-destructive detection framework using hyperspectral imaging to investigate nitrogen–chlorophyll synergy and spatial distributions in greenhouse tomatoes under nitrogen stress. Our integrated approach covered pattern exploration, mechanism interpretation, model development, and practical application for precision nitrogen management. The key findings reveal distinct spatiotemporal patterns: nitrogen and chlorophyll contents peaked at the flowering stage and showed a consistent vertical distribution (upper > middle > lower leaves). We established accurate quantitative detection models through a three-stage wavelength selection strategy combining seven algorithms. The nitrogen detection model (iRF–CARS–IRIV–PLSR) achieved an excellent performance (R = 0.8676, RMSE = 0.6284%, RPD = 2.0110) using only 28 key wavelengths. Similarly, the chlorophyll model (iVISSA–CARS–GA–PLSR) demonstrated a strong predictive ability (R = 0.8741, RMSE = 0.2503 mg/g, RPD = 2.0587) with 33 wavelengths. By integrating detection models with image processing, we successfully visualized nitrogen and chlorophyll distribution patterns across growth stages and canopy positions. These findings provide a scientific basis for optimizing fertilization strategies and contribute to sustainable greenhouse tomato production.

The proposed coarse–fine–optimal wavelength selection strategy effectively reduces data dimensionality and enhances model generalizability. Future efforts should prioritize model optimization, technology integration, and mechanistic exploration. This includes developing deep forest-based ensemble frameworks to overcome linear modeling limitations, establishing a satellite–UAV–ground collaborative sensing network for multi-scale data fusion, and incorporating plant physiological models to interpret spectral–biological linkages. These advances will support the creation of a multi-modal nutrient diagnostic framework and portable devices, improving the accuracy and applicability of precision horticulture.

## Figures and Tables

**Figure 1 plants-14-03276-f001:**
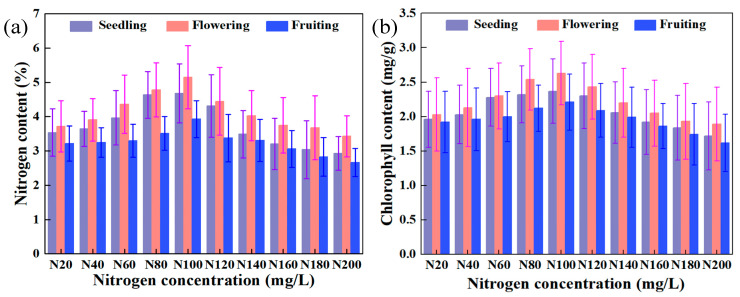
Dynamics and differences in the content of nitrogen and chlorophyll in tomato leaves influenced by nitrogen concentration in the nutrient solution and growth stage. (**a**) Nitrogen content; (**b**) chlorophyll content.

**Figure 2 plants-14-03276-f002:**
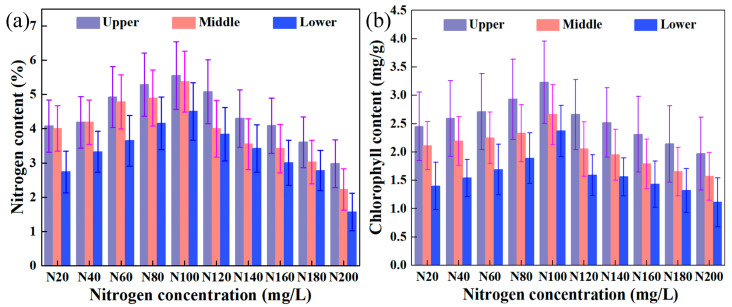
Dynamics and differences in the content of nitrogen and chlorophyll in tomato leaves influenced by nitrogen concentration in the nutrient solution and leaf spatial position. (**a**) Nitrogen content; (**b**) chlorophyll content.

**Figure 3 plants-14-03276-f003:**
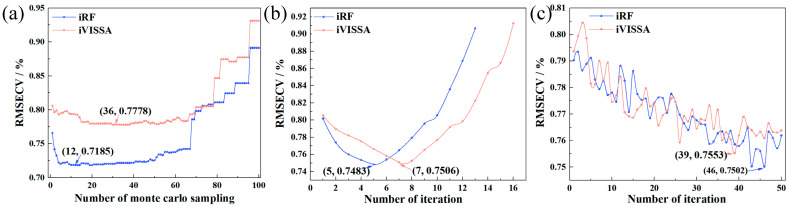
The RMSE_CV_ value of extraction process of nitrogen. (**a**) CARS; (**b**) BOSS; (**c**) VCPA. (Note: iRF: Interval Random Forest; iVISSA: Interval Variable Iterative Space Shrinkage Approach; CARS: Competitive Adaptive Reweighted Sampling; BOSS: Bootstrapping Soft Shrinkage; VCPA: Variable Combination Population Analysis; IRIV: Iteratively Retaining Informative Variables; GA: Genetic Algorithm. iRF and iVISSA perform coarse extraction of key wavelengths).

**Figure 4 plants-14-03276-f004:**
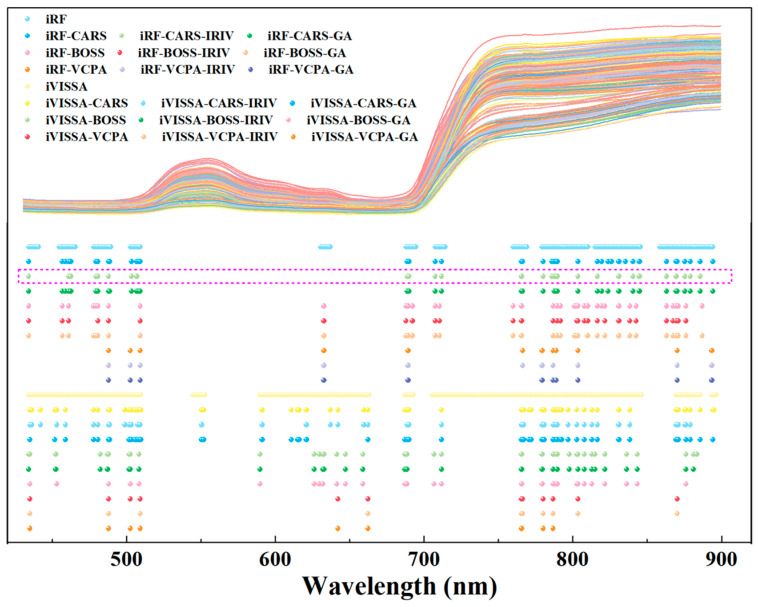
Distribution of key wavelengths resulting from the three-stage feature extraction strategies for nitrogen, ranging from coarse to coarse–fine to coarse–fine–optimal, in tomato leaves. (Note: iRF and iVISSA perform coarse extraction of key wavelengths. CARS, BOSS, and VCPA conduct refined selection from the coarse-extracted wavelengths. IRIV and GA further optimize the selected wavelength sets. The points in the figure represent the distribution of key wavelengths selected by each algorithm for nitrogen detection. The subset highlighted by the pink box indicates the final optimal set of key wavelengths identified through the complete screening process).

**Figure 5 plants-14-03276-f005:**
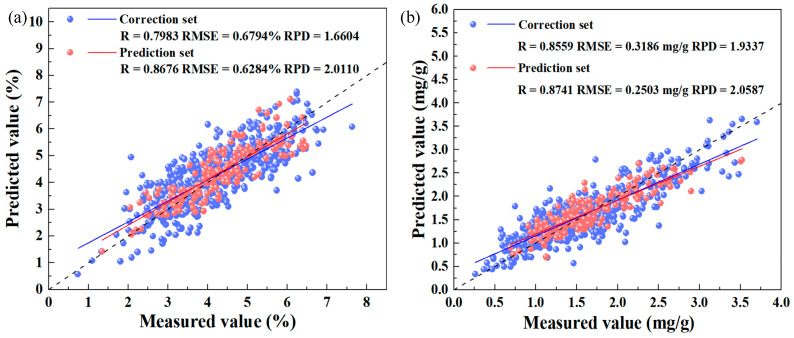
Model performance for nitrogen and chlorophyll content in tomato leaves across calibration and prediction sets. (**a**) Nitrogen, fitted using the iRF–CARS–IRIV–PLSR model; (**b**) chlorophyll, fitted using the iVISSA–CARS–GA–PLSR model.

**Figure 6 plants-14-03276-f006:**
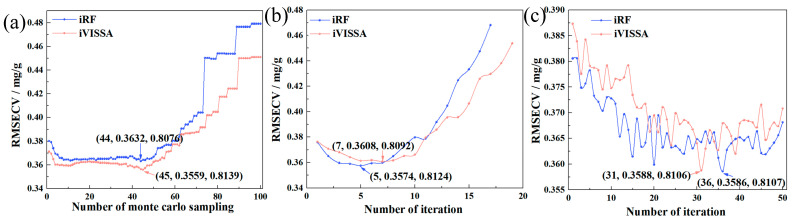
The RMSE_CV_ value of extraction process of chlorophyll. (**a**) CARS; (**b**) BOSS; (**c**) VCPA.

**Figure 7 plants-14-03276-f007:**
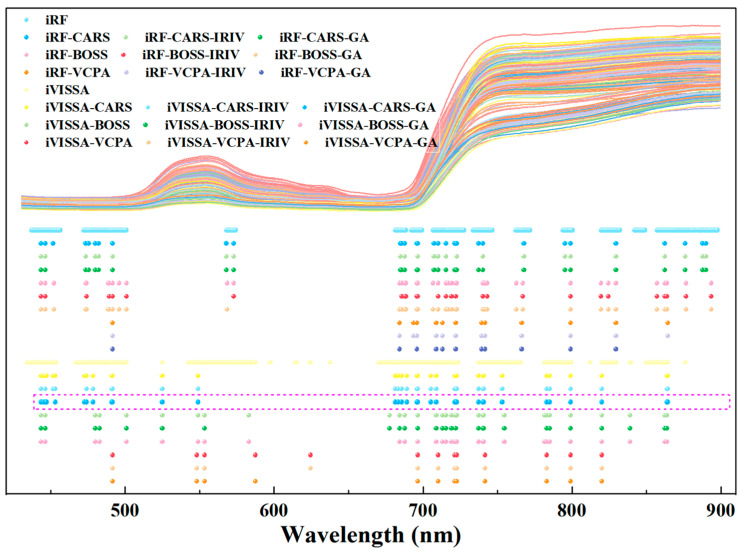
Distribution of key wavelengths resulting from the three-stage feature extraction strategies for chlorophyll, ranging from coarse to coarse–fine to coarse–fine–optimal, in tomato leaves. (Note: The points in the figure represent the distribution of key wavelengths selected by each algorithm for chlorophyll detection. The subset highlighted by the pink box indicates the final optimal set of key wavelengths identified through the complete screening process).

**Figure 8 plants-14-03276-f008:**
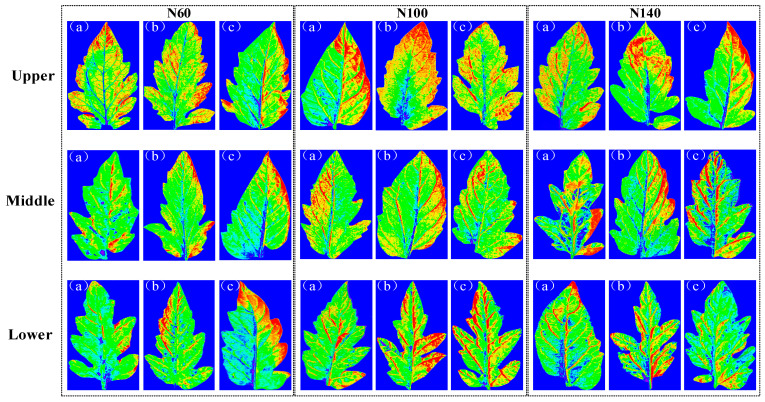
The visual results of the content and distribution of the nitrogen for the leaves. (**a**–**c**) show seedling, flowering, and fruiting stages, respectively. (Note: The quantitative gradient color scheme uses blue to represent 0% nitrogen content and red for 7.64%. The color coding progresses through transitional hues (such as light blue, green, yellow, and orange) to represent intermediate nitrogen values, with higher values shifting toward red and lower values toward blue).

**Figure 9 plants-14-03276-f009:**
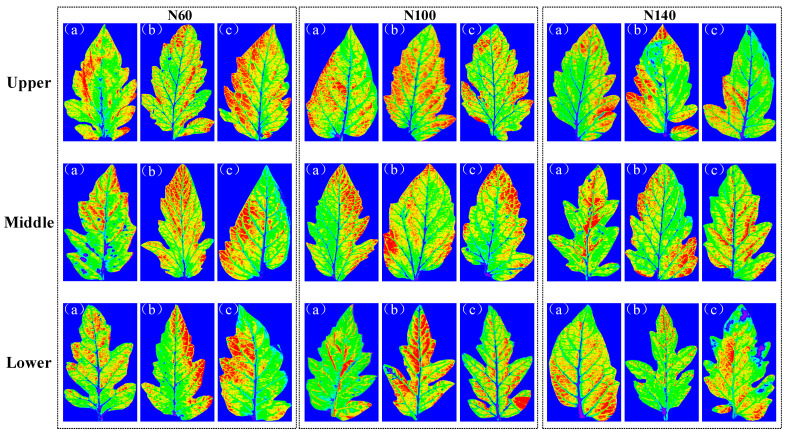
The visual results of the content and distribution of the chlorophyll for the leaves. (**a**–**c**) show seedling, flowering, and fruiting stages, respectively. (Note: The quantitative gradient color scheme uses blue to represent 0 chlorophyll content and red for 3.70 mg/g. The color coding progresses through transitional hues (such as light blue, green, yellow, and orange) to represent intermediate chlorophyll values, with higher values shifting toward red and lower values toward blue).

**Figure 10 plants-14-03276-f010:**
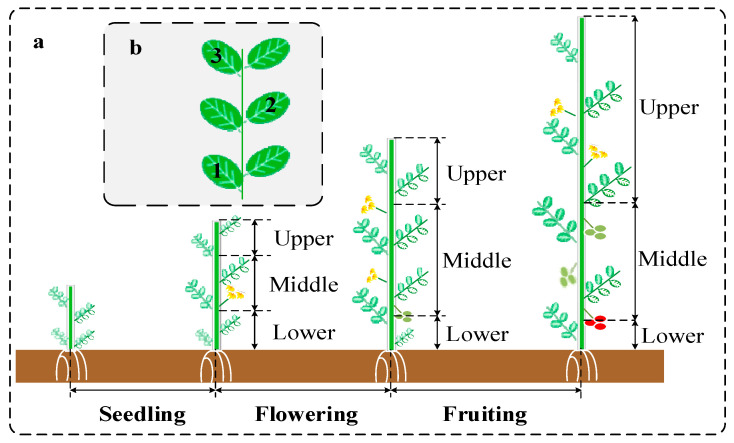
Schematic diagram of the stratified structure and leaf sampling in tomato plants. (**a**) Illustration of the upper, middle, and lower leaf layers at different growth stages. (**b**) Diagram of leaf sampling on a branch (collected leaves are marked).

**Table 1 plants-14-03276-t001:** Prediction results of the nitrogen content of tomato leaves by PLSR.

Coarse	Fine	Optimization	NV ^1^	Calibration			Prediction		
				R_C_	RMSE_C_	RPD_C_	R_P_	RMSE_P_	RPD_P_
**iRF**	-	-	231	0.7434	0.7383	1.4951	0.8153	0.6481	1.7274
	**CARS**	-	42	0.7637	0.7041	1.5490	0.8409	0.6367	1.8478
		**IRIV**	**28**	**0.7983**	**0.6794**	**1.6604**	**0.8676**	**0.6287**	**2.0110**
		GA	41	0.7767	0.6953	1.5876	0.8433	0.6323	1.8607
	BOSS	-	36	0.7541	0.7245	1.5226	0.8385	0.6358	1.8352
		IRIV	30	0.7763	0.7197	1.5864	0.8467	0.6326	1.8794
		GA	34	0.7688	0.7114	1.5637	0.8304	0.6289	1.7948
	VCPA	-	12	0.8351	0.6430	1.8179	0.8177	0.6718	1.7372
		IRIV	11	0.8639	0.6237	1.9855	0.8558	0.6281	1.9331
		GA	11	0.8498	0.6302	1.8972	0.8419	0.6349	1.8531
iVISSA	-	-	451	0.7042	0.7879	1.4084	0.7464	0.7304	1.5026
	CARS	-	66	0.7117	0.7770	1.4235	0.7697	0.7109	1.5664
		IRIV	47	0.7485	0.7346	1.5080	0.7882	0.6738	1.6249
		GA	55	0.7103	0.8085	1.4207	0.7727	0.6572	1.5754
	BOSS	-	38	0.7441	0.7374	1.4969	0.7534	0.7258	1.5208
		IRIV	31	0.7546	0.7224	1.5240	0.7967	0.6647	1.6546
		GA	30	0.7451	0.7302	1.4994	0.7669	0.7198	1.5582
	VCPA	-	12	0.8277	0.6614	1.7820	0.7651	0.7215	1.5530
		IRIV	10	0.8363	0.6573	1.8239	0.7798	0.7018	1.5974
		GA	9	0.8316	0.6501	1.8006	0.7704	0.7164	1.5685

^1^ Number of variables. Bold indicates the optimal result.

**Table 2 plants-14-03276-t002:** Prediction results of the chlorophyll content of tomato leaves by PLSR.

Coarse	Fine	Optimization	NV	Calibration			Prediction		
				R_C_	RMSE_C_	RPD_C_	R_P_	RMSE_P_	RPD_P_
**iRF**	-	-	280	0.8028	0.1117	1.6772	0.8430	0.0820	1.8590
	CARS	-	34	0.8188	0.1102	1.7419	0.8515	0.0794	1.9071
		IRIVs	27	0.8391	0.1016	1.8383	0.8665	0.0705	2.0033
		GA	29	0.8302	0.1053	1.7938	0.8598	0.0765	1.9584
	**BOSS**	-	27	0.8381	0.1029	1.8331	0.8456	0.0808	1.8733
		**IRIVs**	**21**	**0.8577**	**0.0927**	**1.9450**	**0.8723**	**0.0645**	**2.0451**
		GA	24	0.8452	0.0993	1.8711	0.8597	0.0765	1.9577
	VCPA	-	14	0.8036	0.1112	1.6802	0.8452	0.0812	1.8711
		IRIVs	12	0.8104	0.1106	1.7068	0.8483	0.0798	1.8885
		GA	14	0.8036	0.1112	1.6802	0.8452	0.0812	1.8711
iVISSA	-	-	475	0.7949	0.1132	1.6482	0.8419	0.0846	1.8531
	CARS	-	49	0.8114	0.1109	1.7109	0.8429	0.0843	1.8585
		IRIVs	36	0.8244	0.1062	1.7668	0.8500	0.0805	1.8983
		GA	46	0.8186	0.1102	1.7411	0.8505	0.0803	1.9012
	BOSS	-	36	0.8325	0.1047	1.8050	0.8458	0.0800	1.8744
		IRIVs	32	0.8412	0.1004	1.8494	0.8543	0.0767	1.9239
		GA	34	0.8411	0.1004	1.8488	0.8484	0.0798	1.8891
	VCPA	-	12	0.8054	0.1110	1.6871	0.8472	0.0806	1.8937
		IRIVs	10	0.8183	0.1105	1.7398	0.8504	0.0802	1.9006
		GA	11	0.8137	0.1075	1.7203	0.8499	0.8005	1.8977

Bold indicates the optimal result.

**Table 3 plants-14-03276-t003:** The fertilizer content (mg/L), EC, and pH under different nitrogen concentrations.

Fertilizer	N20	N40	N60	N80	N100	N120	N140	N160	N180	N200
Calcium nitrate	0	307.48	605.68	913.15	1216	1216	1216	1216	1216	1216
Chelated calcium fertilizer	491.57	367.27	246.72	122.43	0	0	0	0	0	0
Urea	0	0	0	0	0	131.67	262.34	395.01	526.68	658.35
Calcium ammonium nitrate	42.1	42.1	42.1	42.1	42.1	42.1	42.1	42.1	42.1	42.1
Potassium nitrate	395	395	395	395	395	395	395	395	395	395
Potassium dihydrogen phosphate	208	208	208	208	208	208	208	208	208	208
Potassium sulfate	393	393	393	393	393	393	393	393	393	393
Magnesium sulfate	466	466	466	466	466	466	466	466	466	466
Total nitrogen concentration	59.64	121.14	181.7	242.27	302.84	363.41	423.98	484.54	545.54	605.68
EC	2.49	2.42	2.40	2.30	2.29	2.33	2.43	2.50	2.53	2.56
pH	6.88	6.84	6.80	6.95	6.99	7.05	7.00	6.95	6.96	6.98

**Table 4 plants-14-03276-t004:** Dataset partitioning results for nitrogen content (%) and chlorophyll content (mg/g) in leaves.

Target	Dataset	Number of Sample	Maximum	Minimum	Mean	Standard Deviation
Nitrogen	Calibration set	551	7.6357	0.7374	4.1669	1.0721
	Prediction set	184	6.4984	2.0761	4.2257	0.9627
Chlorophyll	Calibration set	551	3.7026	0.2614	1.6102	0.6099
	Prediction set	184	3.5208	0.6898	1.6795	0.5228

## Data Availability

The dataset is available on request from the authors.

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
