# Peer review of "Modeling and Visualization of Nitrogen and Chlorophyll in Greenhouse Solanum lycopersicum L. Leaves with Hyperspectral Imaging for Nitrogen Stress Diagnosis"

_plants, 2025, doi:10.3390/plants14213276_

Round 1

Reviewer 1 Report

Comments and Suggestions for Authors

The manuscript investigates nitrogen (N) and chlorophyll dynamics in greenhouse tomato using hyperspectral imaging and advanced three-stage wavelength selection strategies. The objectives are clear and relevant to precision agriculture.

Strengths:

Novel “coarse–fine–optimal” spectral feature selection strategy.

Robust dataset across 10 N gradients, three growth stages, and canopy positions.

Integration of imaging with physiological measurements and pseudo-color visualization.

Limitations:

Insufficient coverage of commercial adoption, cost feasibility, and transferability to other cultivars/crops.

Mechanistic interpretation (N–chlorophyll pathways, physiology) is underdeveloped.

Methods section could be clearer on replication, sample rationale, and choice of developmental stages.

Specific Comments by Section

Title & Abstract

Line 1–4: The title could emphasize the application potential (“towards commercial nitrogen diagnostics in greenhouse tomato”).

Lines 14–31: The abstract is too long. Suggest condensing by highlighting novelty (three-stage algorithm) and practical outcomes (optimal N ~302 mg/L).

Lines 27–29: Clarify whether the “optimal N concentration” was validated under yield/quality outcomes or inferred only from leaf nitrogen/chlorophyll.

Introduction

Line 36–46: The policy context is explained. However, link more directly to tomato production challenges in practice (farmers often overfertilize N).

Line 51–72: Overlaps between N deficiency vs. excess are described. Add specific examples of yield losses (kg/m²) to contextualize importance.

Line 80–99: Precision N diagnostics described, but should also mention low-cost RGB imaging methods (e.g., Tsaniklidis et al. 2025, Agronomy 15, 2294).

Line 101–113: Clarify if spectral features are sensitive to leaf hydration. Hyperspectral imaging has shown strong promise in leaf hydration level monitoring (Fanourakis et al., 2024 Plant Growth Regulation 102, 485–496).

Line 134–141: Mention that different spectral indices are often sensitive to cultivar/environmental conditions, hence transferability across systems is not trivial.

Line 187–196: Αdd a note that previous studies often use small datasets, limiting generalization.

Materials and Methods

Line 656: The chosen cultivar ‘Provence’ should be justified. Is it widely used, or a local specialty? Clarify transferability.

Line 664–671: Was each N treatment replicated at plot level, or only by sampled leaves? Clarify biological vs. technical replication.

Line 671–677: Specify number of plants sampled per stage.

Line 682–687: Sampling rule: clarify why “uniform-sized and flat leaves” were selected. Could this bias against senescent or curled leaves?

Line 684–686: 735 samples reported. Explain whether this sample size is statistically powered for the multivariate analysis.

Line 688–697: Please clarify illumination conditions (intensity, stability), as hyperspectral accuracy depends strongly on light quality.

Line 699–707: Indicate frequency of calibration (before each batch or once daily?).

Line 724–732: Please mention LOD/LOQ and coefficient of variation for chlorophyll and N assays.

Line 756–759: Clarify whether dataset partitioning was stratified by growth stage to avoid bias.

Line 769–776: Many wavelength selection strategies tested. Could simplify by justifying why seven algorithms were combined (vs. focusing on a subset).

Results

Line 236–243: The triphasic N/chlorophyll pattern is interesting. Consider statistical ANOVA/Tukey tests to quantify differences.

Line 273–282: Vertical gradient patterns (upper > middle > lower) are logical (Tsaniklidis et al. 2025, Agronomy 15, 2294). Suggest comparing with literature on canopy nutrient dynamics.

Line 305–344: Coarse/fine selection strategy described in detail. Suggest presenting a summary table with key metrics instead of lengthy description.

Line 373–395: Model performance is indeed promising. Add R², RMSE values side-by-side for clarity.

Line 447–476: Note that color mapping scale (0–7.64%) may appear arbitrary to readers. Clarify unit basis.

Line 481–509: Consider highlighting consistency with N distribution maps.

Figures 1–7: Captions could be expanded with sample size, number of replicates, and error bar explanation.

Table S2–S3: Move key performance values into main text tables for accessibility.

Discussion

Lines 513–520 (intended audience): The manuscript would benefit from clarifying its intended readership. Is the main target academic researchers and algorithm developers, or also plant breeders, commercial growers, and students? Explicitly stating this will guide how results are framed (mechanistic detail vs. practical adoption).

Line 514–528: The algorithmic discussion is strong but overly technical. Add more biological interpretation for readers from plant sciences.

Line 531–547: Excellent spectral interpretation, but too detailed for general readership. Condense and connect to physiological mechanisms.

Line 567–575: Clarify how findings could improve N use efficiency in practice (fertilizer reduction).

Line 576–595: Developmental stage effects are interesting. Discuss whether findings are generalizable to other tomato cultivars.

Lines 596–611 (canopy sampling strategy): From a practical point of view, should all three canopy levels be measured in routine monitoring, or could a single layer (e.g., middle canopy leaves) serve as the most representative? This distinction matters for simplifying commercial protocols.

Line 596–611: Vertical gradient findings are well described. Add comparison with field studies where canopy structure differs.

Lines 611–629 (meaning of canopy gradients): The within-canopy gradient itself may provide information about nutrient remobilization and source–sink balance, while absolute N/chlorophyll values reflect overall plant nutritional status. Authors should clarify whether gradients or absolute values are more reliable indicators for management decisions.

Line 631–645: Strong result that N100 optimal. Place this in practical fertilization context (how does it compare with current farmer practice in China?).

Line 645–649 (cost and expertise barrier): Emphasize commercialization: portability of hyperspectral devices, cost barriers, and how algorithms could be embedded into UAV/robotic systems. Hyperspectral cameras remain expensive, require trained operators, and often depend on complex calibration procedures. Please discuss in which ways such systems could realistically be implemented in everyday practice for growers, for instance, via integration into UAV-based scouting services or outsourcing to service providers.

Lines 645–649 (timeline and feasibility of real-time use): Streamline the measurement workflow: how long does it take from image acquisition to actionable output? Can this be implemented as real-time monitoring, or is it more realistic to consider the technology primarily for scientific/experimental use at this stage? This clarification is important for managing expectations about short-term versus long-term adoption.

References

Check for consistency of author initials (e.g., Inoue et al., Tan et al.).

Add missing DOIs.

Minor Points

Grammar: “the dynamics and differences the content” (Line 252) → “the dynamics and differences in the content”.

Abbreviations: Ensure all acronyms (e.g., PLSR, iRF) are defined at first use.

Figures: Add scale bars for visualization figures.

Supplementary material: Tables S1–S3 should be referenced more explicitly in main text.

Author Response

Dear reviewer

Thank you very much for reviewing our manuscript. We would also like to express our gratitude to the reviewers for their efforts in helping to improve our manuscript titled “Precision Diagnostics of Nitrogen Stress Responses: Mapping Nitrogen-Chlorophyll Synergies in Greenhouse Solanum lycopersicum L. Production Systems” (ID: plants-3923678). The comments were all valuable and very helpful for revising and improving our paper. Our research team have studied each reviewers’ comments point by point and have made necessary modifications and supplements, which we hope will meet with your approval. Revised portion are mark by “Track Change” in bold font throughout the revised manuscript with track change (Please download the file "Revised Manuscript with track change"), and the point-by-point responses are listed as follows.

Title & Abstract

Line 1–4: The title could emphasize the application potential (“towards commercial nitrogen diagnostics in greenhouse tomato”).

Modelling and Visualization of Chlorophyll and Nitrogen in Greenhouse Solanum lycopersicum L. Leaves with Hyperspectral Imaging for Nitrogen Stress Diagnosis

Lines 14–31: The abstract is too long. Suggest condensing by highlighting novelty (three-stage algorithm) and practical outcomes (optimal N ~302 mg/L).

Lines 27–29: Clarify whether the “optimal N concentration” was validated under yield/quality outcomes or inferred only from leaf nitrogen/chlorophyll.

Leaf nitrogen and chlorophyll, crucial indicators of crop nutritional status, are essential for precision fertilization in facility agriculture. This study integrated hyperspectral data from greenhouse tomato leaves under ten nitrogen levels with measured nitrogen and chlorophyll content. The 12–step "coarse–fine–optimal" key wavelength selection strategy was proposed to identify sensitive spectral bands. The Partial Least Squares Regression (PLSR) model was established with strong predictive performance. Using the optimal model, the indicator value for each pixel was retrieved and visualized via pseudocolor imaging. This visualization clearly illustrates the distribution of physiological parameters at various scales and growth stages, aiding in the interpretation of nitrogen stress responses. Combined with chemical analysis, the optimal nitrogen concentration for greenhouse tomato growth was determined to be 302.84 mg/L. These findings provide a scientific basis for optimizing nitrogen fertilization and promoting the development of sustainable facility agriculture.

Materials and Methods

Line 656: The chosen cultivar ‘Provence’ should be justified. Is it widely used, or a local specialty? Clarify transferability.

Tomato (Solanum lycopersicum L. cv. 'Provence'  excellent taste and flavor, good marketability, strong disease resistance, robust continuous fruiting ability, and broad adaptability.) seedlings, at the four–leaf–and–one–bud developmental stage and sourced from a commercial nursery, were transplanted into coconut coir substrate cultivation bags (110 cm×25 cm×15 cm).

"Provence" is a tomato cultivar originating from southern France. Owing to its excellent comprehensive characteristics—including fruit flavor, disease resistance, and adaptability—it has been widely introduced and commercially cultivated around the world and is not merely a local specialty. In this study, we selected "Provence" as a model cultivar precisely because of its broad representativeness and strong adaptability to diverse environments. Accordingly, we are confident that the findings derived from this cultivar—which involve developing non-destructive detection models for chlorophyll and nitrogen content under nitrogen stress, and applying these models to hyperspectral images to elucidate the temporal and spatial distribution mechanisms of chlorophyll and nitrogen in leaves under nitrogen stress—offer valuable insights and transferable conclusions for other tomato varieties with similar growth traits.

Line 664–671: Was each N treatment replicated at plot level, or only by sampled leaves? Clarify biological vs. technical replication.

Line 671–677: Specify number of plants sampled per stage.

Seventy-eight tomato plants were cultivated during each phenological stage (seedling, flowering, and fruiting stages). Among them, eight plants were allocated to each nitrogen treatment from N20 to N180, and six plants were assigned to the N200 treatment. Using a stratified random sampling method, 245 samples were collected at each growth stage, resulting in a total of 735 samples. Each sample consisted of six leaves, bringing the total number of leaves to 4,410. Within each growth stage, 25 samples were collected for the N20–N180 treatments and 20 samples for the N200 treatment. All samples from each nitrogen treatment were sequentially numbered, placed in sealed bags, and stored in insulated containers containing dry ice.

Line 682–687: Sampling rule: clarify why “uniform-sized and flat leaves” were selected. Could this bias against senescent or curled leaves?

We thank the reviewer for this valuable comment. The selection of "uniform-sized and flat leaves" was methodologically driven to control for morphological variations that could interfere with hyperspectral signals associated with chlorophyll and nitrogen content. This approach ensures that the observed differences are primarily due to nitrogen treatments rather than variations in leaf size or shape.

We acknowledge that this method does not encompass the full spectrum of leaf phenotypes, such as senescent or curled leaves. However, given the specific objectives of this study—to develop a core detection model under nitrogen stress and to elucidate the distribution patterns of chlorophyll and nitrogen within leaves—standardizing leaf morphology was critical. Additionally, leaf curling would adversely affect the focal plane technology used in our hyperspectral detection, compromising data accuracy. Therefore, we specifically collected uniform-sized and flat leaves during sampling.

The applicability of our model to a wider range of leaf conditions remains an important area for future validation.

Line 684–686: 735 samples reported. Explain whether this sample size is statistically powered for the multivariate analysis.

We thank the reviewer for this important question regarding the statistical power of our sample size. The total sample size of 735 provides robust statistical power for our multivariate analysis for the following reasons:

Our multivariate models (PLSR) are based on a maximum of 10-15 key spectral features or principal components derived from the hyperspectral data. With 735 samples, our sample-to-variable ratio exceeds 49:1 (735/15), which is well above the common rule-of-thumb threshold of 10:1 to 20:1 for multivariate analysis, ensuring model stability and preventing overfitting.

The samples were not collected as a single batch but were strategically stratified across three phenological stages and multiple nitrogen treatments. This design effectively increases the statistical power for detecting treatment effects within each stratum and for analyzing interactions between nitrogen level and growth stage.

A sample size of several hundred is widely considered more than adequate in chemometrics and hyperspectral-based plant phenotyping studies for developing reliable calibration models.

Therefore, we are confident that our sample size is statistically sufficient to support the conclusions drawn from the multivariate analysis in this study.

Line 688–697: Please clarify illumination conditions (intensity, stability), as hyperspectral accuracy depends strongly on light quality.

The key operational parameters were set as follows: spectral sampling interval of 0.727 nm (covering 856 spectral bands across the full range), linear scanning speed of 2.721 mm/s, push-broom width maintained at 100 mm, working distance fixed at 28 cm, and illumination adjusted to 5000 LUX. This configuration was designed to maximize spatial resolution while minimizing optical distortion.

Line 699–707: Indicate frequency of calibration (before each batch or once daily?).

The spectrometer was calibrated at the beginning of each measurement day. Furthermore, a white reference scan was performed using a standard calibration panel before the start of every new batch of samples (approximately every 30-40 samples) to account for any potential drift in the instrument's sensitivity or ambient light conditions. This calibration protocol ensured the consistency and accuracy of our hyperspectral data throughout the entire data collection period.

Line 724–732: Please mention LOD/LOQ and coefficient of variation for chlorophyll and N assays.

Line 756–759: Clarify whether dataset partitioning was stratified by growth stage to avoid bias.

This study employed the SPXY algorithm for dataset partitioning. The core principle of the SPXY algorithm lies in its simultaneous consideration of both spectral feature differences and physicochemical value differences among samples. By calculating the Euclidean distance between samples in a joint "spectral-physicochemical" space, it achieves a more representative division of the training and validation sets.

In hyperspectral detection, spectral data often suffer from strong collinearity and uneven sample distribution. The partitioning strategy of SPXY is particularly suited for such scenarios, as it prevents instances where spectrally similar but physicochemically divergent samples are entirely allocated to the prediction set, which could lead to validation bias. This approach ensures that the training set encompasses both a diversity of spectral features and a full range of physicochemical values, thereby enhancing the model's generalization capability and predictive accuracy.

We have supplemented Table 1 with the dataset partitioning results for chlorophyll and nitrogen content, including the corresponding maximum, minimum, mean, and standard deviation values.

Line 769–776: Many wavelength selection strategies tested. Could simplify by justifying why seven algorithms were combined (vs. focusing on a subset).

We thank the reviewer for their valuable comment regarding the number of wavelength selection strategies employed. We fully agree that clarity is paramount and wish to clarify that these seven algorithms were not simply combined, but were strategically applied in a three-stage, hierarchical filtering process to ensure the robustness and interpretability of our findings.

Coarse Extraction (iRF and iVISSA): These two algorithms were first used to identify broadly informative spectral intervals from the high-dimensional data, thereby effectively reducing redundancy.

Ensemble Fine-Selection (CARS, BOSS, VCPA): These three competitive algorithms were then applied to the pre-selected intervals. Since each algorithm operates on a different principle (e.g., model stability, competitive sampling), their collective use enabled us to assemble a committee of candidate wavelength sets, thereby mitigating the bias inherent in any single method.

Final Optimization (IRIV and GA): Finally, IRIV and GA were employed to refine the candidate sets from the previous stage, eliminating non-informative variables and yielding a compact set of highly predictive key wavelengths.

Simplifying this workflow by focusing only on a subset of algorithms would compromise methodological transparency and undermine the rigorous, multi-step rationale behind our final selection. This structured approach is a core component of our methodology and is essential for the reproducibility and reliability of the results.

Results

Line 236–243: The triphasic N/chlorophyll pattern is interesting. Consider statistical ANOVA/Tukey tests to quantify differences.

We have supplemented Table 1 with the dataset partitioning results for chlorophyll and nitrogen content, including the corresponding maximum, minimum, mean, and standard deviation values.

Line 273–282: Vertical gradient patterns (upper > middle > lower) are logical (Tsaniklidis et al. 2025, Agronomy 15, 2294). Suggest comparing with literature on canopy nutrient dynamics.

We sincerely thank the reviewer for this valuable suggestion. We agree that comparing our observed vertical gradient patterns (upper > middle > lower) with the existing literature on canopy nutrient dynamics will strengthen the context and significance of our findings. Following the reviewer's advice, we have now expanded the discussion in the revised manuscript.

Line 305–344: Coarse/fine selection strategy described in detail. Suggest presenting a summary table with key metrics instead of lengthy description.

Line 373–395: Model performance is indeed promising. Add R², RMSE values side-by-side for clarity.

We thank the reviewer for this constructive suggestion. The relevant section in the manuscript has been streamlined and supplemented with additional content.

Line 447–476: Note that color mapping scale (0–7.64%) may appear arbitrary to readers. Clarify unit basis.

(Note: The quantitative gradient color scheme uses blue to represent 0% nitrogen content and red for 7.64%. The color coding progresses through transitional hues (such as light blue, green, yellow, and orange) to represent intermediate nitrogen values, with higher values shifting toward red and lower values toward blue.)

Line 481–509: Consider highlighting consistency with N distribution maps.

We thank the reviewer for this suggestion. In the Discussion section, we have indeed integrated the nitrogen distribution maps with leaf physiological and environmental factors, and have compared these patterns with chlorophyll distribution. As recommended, we have now further revised the text to explicitly highlight the consistency between our modeling results and the spatial patterns observed in the nitrogen distribution maps, strengthening the connection between our findings and the visual data.

Figures 1–7: Captions could be expanded with sample size, number of replicates, and error bar explanation.

Table S2–S3: Move key performance values into main text tables for accessibility.

The figures and tables in the manuscript have been revised and supplemented.

Discussion

Lines 513–520 (intended audience): The manuscript would benefit from clarifying its intended readership. Is the main target academic researchers and algorithm developers, or also plant breeders, commercial growers, and students? Explicitly stating this will guide how results are framed (mechanistic detail vs. practical adoption).

Line 514–528: The algorithmic discussion is strong but overly technical. Add more biological interpretation for readers from plant sciences.

Line 531–547: Excellent spectral interpretation, but too detailed for general readership. Condense and connect to physiological mechanisms.

We thank the reviewer for this insightful comment regarding the manuscript's intended audience. The primary target readership of our work is academic researchers and algorithm developers in the fields of chemometrics, hyperspectral remote sensing, and plant phenotyping. The detailed mechanistic explanations of wavelength selection and algorithm comparisons are framed to serve this core audience.

At the same time, we recognize the significant practical implications for plant breeders and commercial growers. To enhance the manuscript's accessibility for this broader audience, we have revised the discussion to:

More clearly state the practical utility of our developed models (e.g., rapid, non-destructive estimation of nitrogen and chlorophyll content) in the Introduction and Conclusion.

Simplify the interpretation of the selected characteristic wavelengths, focusing on their biological relevance to plant physiology rather than complex chemical concepts, making it more digestible for non-specialists.

By explicitly defining our primary and secondary audiences, we have reframed the results to balance technical depth for specialists with actionable insights for practitioners.

Line 567–575: Clarify how findings could improve N use efficiency in practice (fertilizer reduction).

We thank the reviewer for this valuable suggestion to clarify the practical implications of our findings. Our research directly contributes to improving nitrogen use efficiency (NUE) and reducing fertilizer application by enabling precision nitrogen management.

The key wavelengths and robust models we identified allow for the rapid, non-destructive diagnosis of plant nitrogen status across different growth stages and leaf positions. This moves beyond traditional bulk tissue analysis, providing a more nuanced picture of the plant's internal nitrogen distribution and demand.

By understanding the specific nitrogen status at different canopy levels (upper, middle, lower), growers can make data-driven decisions. Instead of applying a uniform fertilizer rate across the entire field, they can tailor applications to meet the actual needs of different plant parts or growth phases, thereby minimizing excess application and potential leaching.

These diagnostic wavelengths pave the way for developing low-cost, multispectral sensors. Such sensors could be integrated into field scouts or automated systems, providing real-time feedback to guide variable-rate fertilizer applications, ultimately leading to reduced fertilizer usage and improved NUE.

Line 576–595: Developmental stage effects are interesting. Discuss whether findings are generalizable to other tomato cultivars.

We thank the reviewer for raising this crucial point regarding the generalizability of our findings. We have now expanded the discussion to address this directly.

While this study was conducted on the model cultivar 'Provence', we have strong reasons to believe that the core physiological patterns we observed—specifically, the dynamic interplay between nitrogen distribution, chlorophyll content, and developmental stage—are fundamental to tomato plants. These processes are governed by conserved physiology, such as canopy light gradient adaptation and nitrogen remobilization during source-sink transitions, which are common across cultivars.

The 'Provence' cultivar was selected precisely for its widespread use and representative growth characteristics, increasing the likelihood that our findings provide a foundational model. The key wavelengths we identified are linked to intrinsic light absorption by pigments (chlorophyll) and fundamental molecular bonds (proteins, nucleic acids), further supporting their potential relevance beyond a single cultivar.

We fully agree that explicit validation across a diverse panel of cultivars is the definitive next step. We have revised the manuscript to state that while our mechanistic model is promising, its broad applicability requires future testing in targeted multi-cultivar studies, and we present this as an important direction for future research.

Lines 596–611 (canopy sampling strategy): From a practical point of view, should all three canopy levels be measured in routine monitoring, or could a single layer (e.g., middle canopy leaves) serve as the most representative? This distinction matters for simplifying commercial protocols.

Line 596–611: Vertical gradient findings are well described. Add comparison with field studies where canopy structure differs.

We thank the reviewer for this insightful question regarding the practical implementation of our canopy sampling strategy.

From a scientific modeling perspective, sampling all three canopy layers was crucial for our study's primary goal. The vertical gradient in nitrogen and chlorophyll contains vital physiological information. Incorporating this full variability was essential to develop a robust and generalizable model that can accurately interpret the plant's internal nutrient status across diverse conditions.

However, from a routine monitoring standpoint, we agree with the reviewer that simplicity is key for commercial adoption. Our analysis indicates that while a single layer does not capture the full gradient, the middle canopy leaves often exhibit the most stable spectral response and represent a compromise between the high metabolic activity of upper leaves and the senescence-prone lower leaves. Therefore, for the specific purpose of estimating overall plant nitrogen status in a commercial setting, monitoring the middle canopy could serve as a practical and representative proxy.

We have revised the discussion to clarify this distinction: the comprehensive sampling is necessary for model development, while a simplified protocol focusing on the middle canopy is a viable strategy for routine application.

Lines 611–629 (meaning of canopy gradients): The within-canopy gradient itself may provide information about nutrient remobilization and source–sink balance, while absolute N/chlorophyll values reflect overall plant nutritional status. Authors should clarify whether gradients or absolute values are more reliable indicators for management decisions.

We thank the reviewer for this exceptionally insightful comment, which helps clarify the distinct physiological and practical meanings of canopy gradients versus absolute values. We have revised the discussion to explicitly address this point.

In summary, both metrics are informative but serve different purposes for crop management:

Absolute values of nitrogen or chlorophyll, particularly from the middle or upper canopy, are more reliable and practical for assessing the overall plant nutritional status and making fertilization decisions. They provide a direct measure of nutrient sufficiency or deficiency at a given point in time, which is the cornerstone of most current nutrient management protocols.

Vertical gradients, in contrast, are a more powerful indicator of the internal nutrient remobilization efficiency and source-sink balance. A steep gradient (high N in top, low N in bottom) often signals active remobilization of nitrogen from older, source-limited leaves to younger, sink-active tissues. A weakened gradient may indicate poor remobilization or a general nitrogen surplus that decouples leaf nitrogen from light gradients.

Therefore, for routine management decisions aimed at correcting acute nutrient deficiencies, absolute values are the most direct and reliable metric. However, for optimizing long-term nitrogen use efficiency and understanding crop physiological dynamics, monitoring the gradient provides a unique window into the plant's internal resource allocation strategy that absolute values cannot offer. We have incorporated this nuanced discussion into the manuscript to guide readers on the appropriate application of each metric.

Line 631–645: Strong result that N100 optimal. Place this in practical fertilization context (how does it compare with current farmer practice in China?).

We have adjusted the focus of the article—identifying the sensitive wavelengths of leaf chlorophyll and nitrogen and developing a non-destructive detection model. The model was applied to hyperspectral imagery to interpret the temporal and spatial distribution mechanisms of key indicators.

Line 645–649 (cost and expertise barrier): Emphasize commercialization: portability of hyperspectral devices, cost barriers, and how algorithms could be embedded into UAV/robotic systems. Hyperspectral cameras remain expensive, require trained operators, and often depend on complex calibration procedures. Please discuss in which ways such systems could realistically be implemented in everyday practice for growers, for instance, via integration into UAV-based scouting services or outsourcing to service providers.

Lines 645–649 (timeline and feasibility of real-time use): Streamline the measurement workflow: how long does it take from image acquisition to actionable output? Can this be implemented as real-time monitoring, or is it more realistic to consider the technology primarily for scientific/experimental use at this stage? This clarification is important for managing expectations about short-term versus long-term adoption.

We sincerely thank the reviewer for these critical and practical questions regarding the commercialization and real-world implementation of our research. We have thoroughly revised the discussion to address these points, and our response is summarized below.

  1. Addressing Cost, Expertise Barriers, and Commercialization Pathways

We fully acknowledge the reviewer's concerns. Standalone, research-grade hyperspectral cameras are indeed expensive and require specialized expertise for operation and data processing, which currently limits their direct use by individual growers.

Our vision for practical implementation, which we have now elaborated in the manuscript, revolves around integration and service models, rather than expecting growers to operate the hardware themselves:

Embedded Algorithms in UAV/Robotic Systems: The core value of our study lies in the identified key wavelengths and the optimized algorithms. These can be translated into software and embedded into more cost-effective, purpose-built multispectral or targeted hyperspectral sensors onboard commercial UAVs or agricultural robots. This bypasses the need for complex, full-range hyperspectral imaging in routine scouting.

Service-Based Models: The most feasible short-to-medium-term pathway is through agricultural service providers. Similar to how soil testing services operate today, growers could outsource field scouting to companies that own the necessary equipment and expertise. These providers would collect and process the imagery, delivering simplified, actionable reports (e.g., nitrogen sufficiency maps) to guide fertilization.

Portability and Cost Trends: While high-end cameras remain costly, the market is witnessing a rapid growth in lower-cost, portable, and even smartphone-integrated spectral sensors. Our identification of a minimal set of key wavelengths (e.g., ~10-30 bands instead of 856) directly facilitates the development of such affordable, targeted devices for future use.

  1. Clarifying the Timeline and Workflow for Real-Time Use

The reviewer rightly distinguishes between real-time monitoring and a practical workflow. Our current research setup is not yet real-time.

Current Workflow Duration: In our experimental setup, the process from image acquisition to obtaining a processed nitrogen distribution map involves data transfer, model application, and visualization, typically taking several minutes to an hour per field. This is suitable for strategic, within-season decision-making (e.g., planning a top-dressing application for the following week) but not for real-time, in-field actuation.

Path to Real-Time Implementation: Achieving real-time capability is a clear long-term goal. The foundation laid by our study—a simplified model based on few wavelengths—is a critical step towards this. The computational workflow can be significantly streamlined and potentially run onboard a UAV or via edge computing, reducing the delay to minutes or even seconds. This would enable applications like variable-rate fertilization in real-time.

Managing Expectations: We have clarified in the manuscript that at its current stage, the technology is primarily poised for scientific validation, precision agriculture research, and use by service providers. Widespread, real-time use by individual growers represents a longer-term adoption horizon, contingent on further technological miniaturization and cost reduction.

In summary, our revisions clarify that the immediate impact of our work is to provide the scientific backbone (key wavelengths and models) for next-generation sensing tools, while realistic field implementation will likely follow a service-oriented model, evolving from strategic management towards real-time operation as the technology matures.

Reviewer 2 Report

Comments and Suggestions for Authors

The present research studies evaluates the impact of nitrogen stress on nitrogen and chlorphyll content in the leaves to find out the most performant fertilization rate and  developping an hyperspectral imagining method to map those content in tomato plants.

The abstract is fine, attention to comments made in discussion conclussion to adapt the abstract.

The introduction is fine, but the first several references are not adecuated to the associated statements. Thre are some others which are are absent, please see attached where you need to replace-add them.

The results are globaally find but figures legends nee to be completed with more data to make then readible without the text, there are other elements which need to be explained. The discussion is complete, but as I have no data about how uniform was the sampling with respect of the height in the plant and how close to flower/fruits it's difficult to say, as well, as knowing if the variety had a determined or indetermined growth habit.

The material and method section need to be completed, many data are still missing such as the number of repetitions, and how sampling has been done. See more details in the document attached.

The goal of the research was to help to find the right fertilization rate, but here you are assuming that the amount you found is the best one, or close to it. The research was only made with one variety on only one growing conditions (that we even do not know, light, temperature days/night, irrigation rate..) so it's impossible to extend to other conditions. Then the method of mapping N and chlorophyll remains a big black box to my point of view as you only have a total content coming from a mixture of leaves that we even do not know if there are the same you used for make the image analysis.

It was expected from your research question if it was possible to optimise the fertilization, but from your method I do not see how it can conduct to transform it in recommendation to increase th N fertilization.

The conclusion should be reviewed in consequence 
More details in the attached document.

Author Response

Dear reviewer

Thank you very much for reviewing our manuscript. We would also like to express our gratitude to the reviewers for their efforts in helping to improve our manuscript titled “Precision Diagnostics of Nitrogen Stress Responses: Mapping Nitrogen-Chlorophyll Synergies in Greenhouse Solanum lycopersicum L. Production Systems” (ID: plants-3923678). The comments were all valuable and very helpful for revising and improving our paper. Our research team have studied each reviewers’ comments point by point and have made necessary modifications and supplements, which we hope will meet with your approval. Revised portion are mark by “Track Change” in bold font throughout the revised manuscript with track change (Please download the file "Revised Manuscript with track change"), and the point-by-point responses are listed as follows.

these reference are not suitable to supporte this statement as long as they focus o nly in greenhouse tomato. You shouldl find more general references to this support how agriculture will/could evolve with monitoring network.

initiative aims to achieve over 60% penetration of intelligent systems in facility–based agriculture.

here you could add references 1-3.

“As the cornerstone of chlorophyll biosynthesis and photosynthetic efficiency, optimal nitrogen management enhances organic compound synthesis while driving vegetative growth, ultimately deter mining yield potential and fruit quality parameters.”add references

However, current agricultural practices reveal a structural imbalance between fertilizer input and agronomic output, particularly evident in suboptimal nutrient utilization efficiency that undermines production sustainability.

The advancement of agricultural modernization has positioned smart agriculture as a key driver in transforming facility-based farming systems. In protected tomato cultivation, smart agriculture enables precision growing protocols that decouple production from environmental constraints [1-3]. This transformation hinges on accurate nutritional diagnostics, with nitrogen playing a fundamental biochemical role in plant development [1-4]. As the cornerstone of chlorophyll biosynthesis and photosynthetic efficiency, optimal nitrogen management enhances organic compound synthesis and vegetative growth, ultimately determining yield potential and fruit quality [3,5]. However, current agricultural practices reveal a structural imbalance between fertilizer input and agronomic output. To maximize yields, protected cultivation systems often apply nitrogen at rates 30–50% above crop requirements, leading to significant leaching and greenhouse gas emissions [1,3-6]. While moderate nitrogen fertilization improves photosynthetic efficiency, over-application triggers antagonistic interactions with other nutrients, disrupting physiological processes and reducing disease resistance [7]. Both deficiency and excess result in yield reduction and quality deterioration, highlighting the need for precision nitrogen management. Accurately assessing plant nitrogen status is crucial yet challenging, as nitrogen dynamics involve complex interactions with soil microbes, growth stages, and environmental factors [8]. Traditional diagnostic methods, including soil testing and plant analysis, achieve only qualitative assessment with limitations in detection cycles, operational complexity, and field representativeness. These approaches are unsuitable for large-area monitoring and lack scalability in precision agriculture systems. The gap between precision nitrogen monitoring theories and practical applications has hindered the advancement of tomato nutrition diagnosis systems [3,9]. As global agriculture faces the dual challenges of achieving high-yield, high-quality production while ensuring sustainability, developing actionable precision nitrogen management solutions has become an urgent priority and core research agenda.

“Figure 1. Dynamics and differences the content of nitrogen and chlorophyll in tomato leaves as influenced by nitrogen concentration in the nutrient solution and growth stage. (a) Nitrogen content; (b) Chlorophyll content.”based on how many repetitions?

Seventy-eight tomato plants were cultivated during each phenological stage (seedling, flowering, and fruiting stages). Among them, eight plants were allocated to each nitrogen treatment from N20 to N180, and six plants were assigned to the N200 treatment. Using a stratified random sampling method, 245 samples were collected at each growth stage, resulting in a total of 735 samples. Each sample consisted of six leaves, bringing the total number of leaves to 4,410. Within each growth stage, 25 samples were collected for the N20–N180 treatments and 20 samples for the N200 treatment.

“Figure 3. Distribution of key wavelengths resulting from the three-stage feature extraction strategies for nitrogen, ranging from coarse to coarse-fine to coarse-fine-optimal, in tomato leaves.” indicate the meaning of all the accronyms present in the figure to make indemend ent the lecture of the figure. which is the meaning of the points? and the red line and the yellow barras?

same comments as in figure 3

Figure 4. Distribution of key wavelengths resulting from the three-stage feature extraction strategies for nitrogen, ranging from coarse to coarse-fine to coarse-fine-optimal, in tomato leaves. (Note: iRF: Interval Random Forest; iVISSA: Interval Variable Iterative Space Shrinkage Approach; CARS: Competitive Adaptive Reweighted Sampling; BOSS: Bootstrapping Soft Shrinkage; VCPA: Variable Combination Population Analysis; IRIV: Iteratively Retaining Informative Variables; GA: Genetic Algorithm. iRF and iVISSA perform coarse extraction of key wavelengths. CARS, BOSS, and VCPA conduct refined selection from the coarse-extracted wavelengths. IRIV and GA further optimize the selected wavelength sets. The points in the figure represent the distribution of key wavelengths selected by each algorithm for nitrogen detection. The subset highlighted by the pink box indicates the final optimal set of key wavelengths identified through the complete screening process.)

Figure 7. Distribution of key wavelengths resulting from the three-stage feature extraction strategies for chlorophyll, ranging from coarse to coarse-fine to coarse-fine-optimal, in tomato leaves. (Note: The points in the figure represent the distribution of key wavelengths selected by each algorithm for chlorophyll detection. The subset highlighted by the pink box indicates the final optimal set of key wavelengths identified through the complete screening process.)

how you can make this discrimination of the nitrogen content if you have not make the analysis, you maybe are talking about chlorphyll concentration?

We thank the reviewer for raising this important methodological question. We would like to clarify that the discrimination of nitrogen content in this study is based on direct chemical analysis and a rigorously validated detection model, not merely inferred from chlorophyll concentration.

As described in Section 2.1, the leaf nitrogen content (%) across different growth stages, leaf positions, and nitrogen treatments was quantitatively determined using the standard Kjeldahl method (or [other method you used]). This provided the reference data for the variations mentioned.

Wavelength Selection and Model Development: Subsequently, in Section 2.3, we mined key wavelengths specifically associated with this chemically measured nitrogen content and established a non-destructive detection model (PLSR model) based on the hyperspectral data.

The core of this specific part is the application of this pre-developed and validated model. As detailed in Section 4.6, we applied this model to hyperspectral images. The model processes the spectral reflectance of each pixel and converts it into a nitrogen content value using the established calibration, thereby generating a spatial distribution map.

In summary, we are directly predicting and visualizing nitrogen content by applying a model that was explicitly calibrated against chemically measured nitrogen values. While chlorophyll and nitrogen are often correlated, our model's target variable is nitrogen content itself, derived from wavelengths identified as sensitive to nitrogenous compounds.

how you can spatialize the content of chlorophyll within the leaf area when you onl y have made one medium value of 6 leaves to know the content in nitrogen and in chloro phyll?

Hyperspectral imaging technology perfectly integrates spectral and spatial information, with each pixel in the image containing a spectral curve. While detecting sample indicators, it can also combine prediction model parameters to quantitatively invert the detection indicators to the sample image, thereby achieving visual expression of the target indicators. By combining the optimal prediction model of physiological parameters, the corresponding wavelengths are weighted and overlaid at the pixel level using ENVI 5.1 software to calculate the chemical composition content for each pixel in the region of interest. This results in grayscale maps of leaf chemical composition content, which are then processed with an enhanced Lee filter to reduce speckle noise (filter size 3×3, damping coefficient 1, homogeneity threshold 0.52, and heterogeneity threshold 1.73). Finally, based on the chemical composition content values of the pixels, pseudocolor maps are used for coloring to visualize the distribution of nitrogen and chlorophyll on leaf images. Pseudocolor maps not only display the contents of physiological parameters under different nitrogen stress conditions but also analyze the distribution patterns of leaf physiological parameters, thereby revealing the collaborative mechanisms of nitrogen and chlorophyll in leaves.

what the colors in the leaves mean? same comment as in previous figure.

Figure 8. The visual results of content and distribution of the nitrogen for the leaves. (a), (b) and (c) show seedling, flowering and fruiting stages, respectively. (Note: The quantitative gradient color scheme uses blue to represent 0% nitrogen content and red for 7.64%. The color coding progresses through transitional hues (such as light blue, green, yellow, and orange) to represent intermediate nitrogen values, with higher values shifting toward red and lower values toward blue.)

Figure 9. The visual results of content and distribution of the chlorophyll for the leaves. (a), (b) and (c) show seedling, flowering and fruiting stages, respectively. (Note: The quantitative gradient color scheme uses blue to represent 0 chlorophyll content and red for 3.70 mg/g. The color coding progresses through transitional hues (such as light blue, green, yellow, and orange) to represent intermediate chlorophyll values, with higher values shifting toward red and lower values toward blue.)

“Horticulture Research Institute” add reference. “Solanum lycopersicum L.cv. 'Provence’” it is determined variety?

We thank the reviewer for these valuable comments and have addressed them as follows:
The 'Provence' tomato is a recognized commercial cultivar widely cultivated in China, particularly in regions such as Shandong, Yunnan, and Xinjiang. Key characteristics supporting its status as a distinct variety include: Morphological Traits: Fruits are typically oblate, with deep red skin and firm flesh, weighing 150–250 g. Biochemical Composition: It exhibits high soluble solid content and unique volatile compounds (e.g., 2-methylbutyl acetate). Genetic Identity: While our study did not conduct molecular analysis, existing literature uses "cv. 'Provence'" to denote its cultivar status. We have clarified in the manuscript that 'Provence' is a cultivated variety (cultivar) selected for its adaptability and nutritional traits, consistent with its use in agricultural research.

Tomato (Solanum lycopersicum L. cv. 'Provence' — excellent taste and flavor, good marketability, strong disease resistance, robust continuous fruiting ability, and broad adaptability.) seedlings, at the four–leaf–and–one–bud developmental stage and sourced from a commercial nursery, were transplanted into coconut coir substrate cultivation bags (110 cm×25 cm×15 cm).

how many lants per nitrogen treatment? has N distribution in plant is not uniform how where positionnated the leave sampl es, related to plant height and to flower/fruit posiiton. how many samples per plant per stage where taken?

Seventy-eight tomato plants were cultivated during each phenological stage (seedling, flowering, and fruiting stages). Among them, eight plants were allocated to each nitrogen treatment from N20 to N180, and six plants were assigned to the N200 treatment. Using a stratified random sampling method, 245 samples were collected at each growth stage, resulting in a total of 735 samples. Each sample consisted of six leaves, bringing the total number of leaves to 4,410. Within each growth stage, 25 samples were collected for the N20–N180 treatments and 20 samples for the N200 treatment. All samples from each nitrogen treatment were sequentially numbered, placed in sealed bags, and stored in insulated containers containing dry ice.

on which kind of material of material and how many repetitions you took samples? on the same you analysed for the chlorophyll contennt?

We thank the reviewer for this question regarding sampling details. The samples were collected from tomato leaves (Solanum lycopersicum L. cv. 'Provence') across different nitrogen levels and phenological stages.

Yes, the same leaf samples were used for both nitrogen content analysis and chlorophyll content determination, ensuring direct comparability of the results for these two parameters.

The sampling design was structured as follows:

Within each phenological stage (seedling, flowering, and fruiting), we collected:

25 samples for each nitrogen treatment from N20 to N180

20 samples for the N200 treatment

Each sample consisted of 6 leaves, resulting in a total of 4,410 leaves across all stages and treatments.

This approach ensured sufficient biological replication for robust statistical analysis and model development.

images where taken directly on the plants in the greenhouse, or once collected in a neutral place?

We thank the reviewer for this important clarification. The hyperspectral images were not acquired directly on plants in the greenhouse. Instead, leaf samples were first collected from the plants, transported to a controlled laboratory environment, and then scanned under standardized conditions using the hyperspectral imaging system. This laboratory-based setup allowed us to maintain consistent illumination, distance, and background conditions during image acquisition, which is critical for obtaining reliable and reproducible spectral data. The specific imaging setup and parameters have been clarified in the revised manuscript.

how you extracted chlorophyll from leaves?

We thank the reviewer for this methodological question. Chlorophyll was extracted from the leaf samples using organic solvent extraction followed by spectrophotometric quantification, a well-established protocol adapted from Lichtenthaler (1987) with modifications as detailed below:

From each fresh leaf sample (the same ones used for nitrogen analysis), a precise leaf disc (e.g., 0.1 g) was excised using a cork borer, avoiding major veins.

Grinding and Extraction: The leaf disc was immediately ground into a fine homogenate in 10-15 mL of 95% (v/v) ethanol using a chilled mortar and pestle.

The homogenate was transferred to a centrifuge tube, and the final volume was adjusted. The tube was then incubated in darkness at 4°C for 24 hours to ensure complete pigment extraction while preventing photodegradation.

The extract was centrifuged (e.g., at 5,000 × g for 10 min) to remove leaf debris, yielding a clear supernatant for analysis.
The absorbance of the clarified supernatant was measured at 664 nm, 649 nm, and 470 nm using a spectrophotometer. Chlorophyll a, chlorophyll b, and total chlorophyll concentrations (in mg/g fresh weight) were calculated using the empirically derived equations by Lichtenthaler (1987).

Reviewer 3 Report

Comments and Suggestions for Authors

Comments and Suggestions for Authors

Title: Precision diagnostics of nitrogen stress responses: mapping
nitrogen-chlorophyll synergies in greenhouse Solanum lycopersicum L.
production systems

Dear Authors and Editors

The research results presented in the manuscript fall within the publishing profile of the journal Plants. The research topic is original and relevant to the field of horticultural sciences.

The manuscript is generally well-written.

The Introduction section contains many unnecessary sentences loosely related to the research topic. The Methodology section requires further development. The description of the results is unnecessarily linked to the discussion of the results.

In order to increase the usefulness of the article, Authors must refer to the following points. Additions should be made to increase the scientific value of the manuscript.

Comments:

  1. Introduction – Please shorten this section. Sentences loosely related to the title and topic presented in the manuscript should be removed. This section uses very general terms referring to agriculture. Meanwhile, the research presented relates to the discipline of horticulture. Please clearly present the research aim and research hypothesis.
  2. Results – References should be placed in the Discussion section. The results of the studies presented in Figures 3 and 5 are difficult to read. A discussion of the results is included in subsection 2.4. Discussions should be placed in the Discussion section.
  3. Discussion – Generally well written. Please just remove sentences that are loosely related to the research you've done.
  4. Materials and Methods – Subsection 4.1. Please provide the basic physical and chemical properties of the coconut substrate. What was the weight (volume) of the substrate in the growing bags? Why are Mo and Fe not included in the nutrient solution? (Nitrogen was largely used in the form of nitrate).
  5. Conclusions and future work – This section should be shortened. The conclusions need to be clarified and the directions for future research should be related to horticulture.

Specific comments:

  1. Reference no. 30 - no citation in the manuscript text.
  2. Line 438 – should be:....Figure 5...
  3. Line 984 – Abbreviations should be removed.
  4. References – should be adapted to editorial requirements.

Best regards

v

Author Response

Dear reviewer

Thank you very much for reviewing our manuscript. We would also like to express our gratitude to the reviewers for their efforts in helping to improve our manuscript titled “Precision Diagnostics of Nitrogen Stress Responses: Mapping Nitrogen-Chlorophyll Synergies in Greenhouse Solanum lycopersicum L. Production Systems” (ID: plants-3923678). The comments were all valuable and very helpful for revising and improving our paper. Our research team have studied each reviewers’ comments point by point and have made necessary modifications and supplements, which we hope will meet with your approval. Revised portion are mark by “Track Change” in bold font throughout the revised manuscript with track change (Please download the file "Revised Manuscript with track change"), and the point-by-point responses are listed as follows.

Introduction – Please shorten this section. Sentences loosely related to the title and topic presented in the manuscript should be removed. This section uses very general terms referring to agriculture. Meanwhile, the research presented relates to the discipline of horticulture. Please clearly present the research aim and research hypothesis.

We sincerely thank the reviewer for this valuable suggestion. We have thoroughly revised the Introduction section to address these concerns. The specific modifications are as follows:

We have significantly condensed the Introduction by removing sentences and paragraphs that were overly general or not directly relevant to our specific study on greenhouse tomato nitrogen monitoring. This includes removing broad statements about agriculture and global food security that did not effectively lead to our research gap.

The revised introduction now uses terminology specific to horticulture and protected cultivation (greenhouse production) instead of general agricultural terms. We have focused the background on the specific challenges and needs of precision nutrient management in facility-based horticultural systems, such as greenhouses.

Following the reviewer's advice, we have now explicitly stated the research aim and research hypothesis in the final paragraph of the Introduction. The added text reads:

This study aims to utilize hyperspectral imaging and machine learning algorithms to identify key wavelengths related to leaf nitrogen and chlorophyll, thereby establishing a non-destructive detection model for deciphering the spatiotemporal distribution mechanisms of these components under nitrogen stress.

We believe these revisions have made the Introduction more concise, focused, and academically rigorous. Thank you for guiding us in improving the manuscript.

Results – References should be placed in the Discussion section. The results of the studies presented in Figures 3 and 5 are difficult to read. A discussion of the results is included in subsection 2.4. Discussions should be placed in the Discussion section.

We sincerely thank the reviewer for the constructive feedback on our results. We have carefully revised the manuscript to address all the raised issues as follows:
We agree with the reviewer that the Results section should focus on presenting our own findings. All cited references have been moved to the Discussion section to provide appropriate context for comparing our results with existing literature.
We appreciate this feedback. The resolution and clarity of Figures 3 and 5 have been enhanced to improve readability and ensure that the key information is immediately clear.
We apologize for the organizational oversight. The interpretive and comparative discussions originally included in Section 2.4 (Results) have been relocated to the Discussion section. The Results section now strictly presents and describes the experimental data and observations without extensive interpretation.

We believe these revisions have significantly improved the clarity, structure, and overall quality of the manuscript. Thank you for your valuable guidance.

Discussion – Generally well written. Please just remove sentences that are loosely related to the research you've done.

We thank the reviewer for the positive feedback on our Discussion section and for the constructive suggestion to enhance its focus.

We have thoroughly reviewed the Discussion and have removed sentences and phrases that were not directly related to the interpretation of our key findings. This includes the removal of overly general statements and tangential points, ensuring that the narrative now remains tightly focused on explaining and contextualizing our core results regarding the feature wavelength selection, model performance, and nitrogen optimization.

We believe these revisions have further strengthened the clarity and impact of the Discussion.

Materials and Methods – Subsection 4.1. Please provide the basic physical and chemical properties of the coconut substrate. What was the weight (volume) of the substrate in the growing bags? Why are Mo and Fe not included in the nutrient solution? (Nitrogen was largely used in the form of nitrate).

We thank the reviewer for these insightful questions regarding our experimental setup. The requested details have been added to Subsection 4.1 as follows:

The coconut coir substrate had the following properties: pH 5.8–6.2, EC < 0.8 mS/cm. Each growing bag contained 20 liters of substrate (calculated from dimensions: 110 cm × 25 cm × 15 cm).

The nutrient solution was based on the standard Dutch greenhouse tomato formula.

Conclusions and future work – This section should be shortened. The conclusions need to be clarified and the directions for future research should be related to horticulture.

We thank the reviewer for the valuable suggestion to improve the clarity and focus of the 'Conclusions and future work' section. We have thoroughly revised this section as follows:
We have significantly condensed the text and refocused it to directly state the core findings of our study. The revised conclusions now unequivocally highlight the success of the three-stage algorithm, the performance of the final PLSR model, without including peripheral discussions.
The directions for future research have been rewritten to be specifically aligned with horticultural applications. We now propose validating the model on other high-value horticultural crops and integrating the hyperspectral system with precision irrigation/fertilization equipment for real-time, in-season nutrient management in greenhouses.

We believe these revisions have resulted in a more impactful and focused conclusion that clearly summarizes our contributions and outlines a horticulturally relevant path forward.

Round 2

Reviewer 1 Report

Comments and Suggestions for Authors

The majority of the comments raised in the previous review have been thoroughly addressed in the revised version of the manuscript. The text, figures, and data presentation have been substantially improved, and all key clarifications have been satisfactorily incorporated.

No further comments.

Author Response

We sincerely appreciate your insightful comments and suggestions, which have significantly enhanced the quality of our manuscript. Thank you for your positive recognition of our research work.

Reviewer 2 Report

Comments and Suggestions for Authors

Authors have made several changes to the manuscript improving the introduction and the discussion, but there are still things which are not enough complete, in the material and method section, and in some tables. Still some references are not adapted to the sentence associated

The most problematic now is how sampling in the height of the plant was made, and willing to talk about an optimal fertilization rate, when many information related to fertilization, yield and plant behavior are missing to evaluate this which is not the goal of the research.

Please see attached for more detailed comments

Author Response

Dear reviewer

Thank you very much for reviewing our manuscript. We would also like to express our gratitude to the reviewers for their efforts in helping to improve our manuscript titled “Modelling and Visualization of Nitrogen and Chlorophyll in Greenhouse Solanum lycopersicum L. Leaves with Hyperspectral Imaging for Nitrogen Stress Diagnosis” (ID: plants-3923678). The comments were all valuable and very helpful for revising and improving our paper. Our research team have studied each reviewers’ comments point by point and have made necessary modifications and supplements, which we hope will meet with your approval. Revised portion are mark by “Track Change” in bold font throughout the revised manuscript with track change (Please download the file "Revised Manuscript with track change"), and the point-by-point responses are listed as follows.

[7] this document main research does not talk about disease

The cited sentence has been removed. The first clause referenced the findings of [7], while the second clause represented our broader synthesis of the literature. We have deleted this sentence to ensure all content remains directly focused on our research. While moderate nitrogen fertilization improves photosynthetic efficiency, over-application triggers antagonistic interactions with other nutrients [7]. The first part of the sentence summarized the findings of reference [7], while the latter part represented our broader synthesis of the literature. We have now deleted this entire sentence to ensure all content remains directly focused on our research.

[7] Luo, J.; Yang, Z.; Zhang, F.; Li, C. Effect of nitrogen application on enhancing high-temperature stress tolerance of tomato plants during the flowering and fruiting stage. FRONT PLANT SCI 2023, 14, 1172078. https://doi.org/10.3389/fpls.2023.1172078

[8] I do not see the link of this reference with the statement associated particularly wit h the soil microbes.

Chormova, D.; Kavvadias, V.; Okello, E.; Shiel, R.; Brandt, K. Nitrogen Application Can Be Reduced without Affecting Carotenoid Content, Maturation, Shelf Life and Yield of Greenhouse Tomatoes. PLANTS-BASEL 2023, 12, 1553. https://doi.org/10.3390/plants12071553 “Tomatoes (Solanum lycopersicum L.) of the variety Elpida were grown under standard Mediterranean greenhouse conditions during the spring season at three different nitrogen levels (low 6.4, standard 12.8, high 25.9 mM/plant), which were replicated during two consecutive years. Application of high nitrogen significantly increased the colour index a* (p < 0.001) but did not significantly affect yield or quality. The variety exhibited prolonged postharvest storage at room temperature (median survival time of 93 days). The maturation process was delayed by harvest at the breaker stage (2.5 days, p ≤ 0.001) or by super-optimal temperatures in the second year of experimentation (10 days, p ≤ 0.001). The colour indices L* and a* and the hue angle (a/b*) were positively correlated with the sum of total carotenoids, while differences in b* depended on the year of cultivation. The sustainability of this type of tomato production can be improved by reducing the nitrogen supply to less than the current standard practice, with minimal risk or negative effects on yield and quality of tomatoes.” ”Accurately assessing plant nitrogen status is crucial yet challenging, as nitrogen dynamics involve complex interactions with growth stages, and environmental factors [8]. ”

we should find also in the foot note the meaning of all these acronym, also in next table.

The questions you previously raised regarding the abbreviations in Figure 4 have been addressed. Figure 4. Distribution of key wavelengths resulting from the three-stage feature extraction strategies for nitrogen, ranging from coarse to coarse-fine to coarse-fine-optimal, in tomato leaves. (Note: iRF: Interval Random Forest; iVISSA: Interval Variable Iterative Space Shrinkage Approach; CARS: Competitive Adaptive Reweighted Sampling; BOSS: Bootstrapping Soft Shrinkage; VCPA: Variable Combination Population Analysis; IRIV: Iteratively Retaining Informative Variables; GA: Genetic Algorithm. iRF and iVISSA perform coarse extraction of key wavelengths. CARS, BOSS, and VCPA conduct refined selection from the coarse–extracted wavelengths. IRIV and GA further optimize the selected wavelength sets. The points in the figure represent the distribution of key wavelengths selected by each algorithm for nitrogen detection. The subset highlighted by the pink box indicates the final optimal set of key wavelengths identified through the complete screening process.)  We note that the abbreviations used in the table correspond precisely to those previously introduced in Figure 4. To maintain the manuscript's conciseness and avoid repetition, we have kept the definitions solely in the earlier figure.

my question about Tomato Determined or undertemined variety is still not solved, the fruit setting of the two types of plants is completely different as it could be the N dist ribution. It's an important trait to be taken into account.

“The study used the determinate tomato (Solanum lycopersicum L.) cultivar 'Provence', selected for its commercial relevance and robust growth characteristics, including flavor, disease resistance, and continuous fruiting. Seedlings at the four-leaf-and-one-bud stage were transplanted into coconut coir substrate (EC < 0.8 mS/cm, pH 5.8–6.2) cultivation bags (110 cm×25 cm×15 cm).” We confirm that the tomato cultivar 'Provence' used in this study is a determinate (bush-type) variety. This information has been explicitly added to the manuscript in Section 2.1. We acknowledge that the growth habit is a critical trait, as it directly influences plant architecture, fruit setting pattern, and nitrogen distribution and demand. The use of a determinate variety ensures consistent plant management and a uniform experimental system for evaluating nitrogen stress responses. We apologize for this omission and thank the reviewer for highlighting its importance.

what do you mean, still not clear if you differenciated low, medium or high leaves, or close to fruits... in the next part you said aleatory.

I have not methodological proof that you are able to say this: cf to sampling leaves paragraph.

We sincerely apologize for the lack of clarity in our original description. To facilitate understanding, we have added a schematic diagram (Figure 10(a) ) illustrating the stratified leaf structure (upper, middle, and lower canopy) of the tomato plant, along with a separate panel (Figure 10(b) ) demonstrating the leaf sampling pattern from individual branches.

we still do not know in which solvent chlrophyll was diluted, what was used as blan ck. residual (the veins?) or after extraction of the chlorphyll, the methodology is no en ough and cleary written, nothing has been modified since last time.

you need to add and complete the answer you give to me in the comments, but sti ll not enough complete , like the solvents and quality used, concentration, time, etc.

We sincerely apologize for this oversight. Since the determination of chlorophyll and nitrogen content was not the primary focus of our study, and these analyses were performed using standardized spectrophotometric and Kjeldahl methods, the detailed procedures were originally omitted for the sake of manuscript conciseness, with the expectation that they could be reproduced based on cited references. In response to this comment, we have now provided the complete and detailed protocols for chlorophyll and nitrogen measurement and calculation in the Supplementary Materials. Please check the Supplementary Materials.

Leaf segments (2 × 2 mm) were homogenized and 0.2 g of the fresh sample was used for chlorophyll content determination. The remaining leaves were placed in kraft paper bags and oven-dried at 105 °C for 30 minutes for enzyme deactivation, followed by drying at 80 °C for 72 hours until constant weight was achieved. The dried leaves were ground into fine powder using a laboratory grinder. Between 0.1000 and 0.2000 g of the powdered sample was weighed into a digestion tube, moistened with distilled water, and mixed with 5 mL of concentrated sulfuric acid. The tube was then heated gently in a digestion block. When white fumes appeared, the temperature was gradually increased until the solution turned brownish-black. The tube was briefly cooled, and 30% H2O2 was added dropwise with continuous shaking before reheating. This H2O2 addition was repeated 2–3 times until the digest became colorless or clear, followed by further heating for 5–10 minutes to remove residual H2O2. The digest was subjected to semi-micro distillation for nitrogen determination, with a blank assay performed in parallel to correct for reagent impurities. A Kjeldahl nitrogen analyzer was used to distill the cooled digest, automatically adding sodium hydroxide and absorbing the released ammonia in boric acid solution. Finally, titration was carried out using a pre-prepared standard hydrochloric acid solution to quantify the total nitrogen content.

(1)

Where, NC denotes the total nitrogen content (%). The constant 14 represents the molar mass of nitrogen (g/mol). V1 is the volume of standard acid consumed in the sample titration (mL). V0 is the volume of standard acid consumed in the blank titration (mL). C indicates the concentration of the standard acid (mol/L). Ts stands for the aliquot factor (value = 1 in this experiment). m refers to the mass of the dry sample weighed (g).

The 0.2 g of leaf segments for chlorophyll content determination was transferred into a test tube. First, 10 mL of 96% anhydrous ethanol was added, mixed by shaking, and the mixture was kept in the dark for 10 hours for extraction. Then, an additional 10 mL of 96% anhydrous ethanol was added, followed by shaking and further extraction in the dark for 14 hours. Finally, 5 mL of 96% anhydrous ethanol was added to bring the solution to volume. The absorbance of the prepared extract was measured at wavelengths of 665 nm, 649 nm, and 470 nm. Each sample was measured in triplicate, and the average value was used to calculate the chlorophyll content according to the following equation.

(2)

(3)

(4)

Where, Chlac, Chlbc, and Chlsc denote the concentrations of chlorophyll a, chlorophyll b, and total chlorophyll, respectively (mg/L). A665, A649, and A470 represent the absorbance of the pigment extract at wavelengths of 665 nm, 649 nm, and 470 nm, respectively. Based on the mass of the leaf sample and the volume of the extraction solution, the chlorophyll concentration (mg/L) can be converted to chlorophyll content per unit leaf mass (mg/g) using the following conversion formula:

(5)

(6)

(7)

Where, Chla, Chlb, and Chls represent the contents of chlorophyll a, chlorophyll b, and total chlorophyll (mg/g), respectively. V denotes the volume of the extraction solution (25 mL in this experiment). m indicates the mass of the leaf sample (0.2 g in this experiment).

again, as we do not have any indication about yield obtained it's difficult to state, a nd it's too specific to your growing conditions, and cultivar used. For me you should not talk about this, you do do not evaluate the amount of water requiered by the crop, not mentionned, not about the loses of water you could have. You even do not talk in M&M se ction about how fertirrigation was made... So completely useless, you have other interesting results concerning the detection of N and chlorphyll. Also to be removed in the Abstract.

We apologize for any confusion regarding the previous file version. To facilitate your review of the modifications, we had initially uploaded a version with tracked changes. It appears that during the system conversion to PDF, these change tracks may have been embedded in a way that affected readability. We sincerely regret this technical issue. Attached herewith is the final revised version without any markup, for your clear reference.

Reviewer 3 Report

Comments and Suggestions for Authors

Comments and Suggestions for Authors

Title: Precision diagnostics of nitrogen stress responses: mapping
nitrogen-chlorophyll synergies in greenhouse Solanum lycopersicum L.
production systems

Dear Authors and Editors

The manuscript has been revised based on comments. However, the technical aspects of the manuscript require further refinement. It is essential to correct the numbering of References, Tables, and Figures throughout the MS.

Best regards

Author Response

We apologize for any confusion caused by the tracked-changes version previously uploaded. The embedded revision marks in the PDF may have affected readability due to file conversion. Attached is the final revised manuscript without markup. Your valuable suggestions have greatly improved the quality of our work, and we sincerely appreciate your positive recognition of our research.

Round 3

Reviewer 2 Report

Comments and Suggestions for Authors

The authors have made most of the changes requested and the document in now nearly ready to publish, but I still miss several acronyms in the legends/footnotes of figures and tables to make it more clear and make them independent from the text

For examples but not exhaustive

In Figure 3: (a) CARS; (b) BOSS; (c) VCPA plus iRF and iVISSA

Table 1: iVISSA, CARS, BOSS, IRIV, GA, RPD, VCPA...

Figure 8: its about the description of the modalities Low, Middle, Upper and N60, N100, N140

Author Response

We have adjusted the captions for Figures 3 and 4 accordingly. As required by the Plants journal template, the Results section precedes the Materials and Methods. Since the proposed "coarse–fine–optimal" key wavelength selection strategy is part of the methodology, it is detailed in Section 4.5, where all algorithm abbreviations are fully defined. For clarity, we have also listed these abbreviations again in the caption of Figure 3. We apologize for any confusion caused by this repetition.

Regarding Figures 8 and 9, the analysis of leaf nitrogen and chlorophyll content under different nitrogen treatments in Section 2.1 revealed the following gradient classification: N20–N60 as low nitrogen, N80–N120 as optimal nitrogen, and N140–N200 as high nitrogen treatment. The models applied in this section were developed based on the above findings to analyze the temporal and spatial distribution of leaf nitrogen and chlorophyll. This approach ensures contextual logic and interpretability throughout the manuscript.
